# Watermark Anything with Localized Messages

**Tom Sander**[1,2,*]**, Pierre Fernandez**[1,*] **Alain Durmus**[3]**, Teddy Furon**[3]**, Matthijs Douze**[1]
[1]Meta FAIR   [2]CMAP, École polytechnique   [3]Centre Inria de l'Université de Rennes
*Equal contribution, correspondence at {tomsander,pfz}@meta.com.

## Abstract

Image watermarking methods are not tailored to handle small watermarked areas. This restricts applications in real-world scenarios where parts of the image may come from different sources or have been edited. We introduce a deep-learning model for localized image watermarking, dubbed the Watermark Anything Model (WAM). The WAM embedder imperceptibly modifies the input image, while the extractor segments the received image into watermarked and non-watermarked areas and recovers one or several hidden messages from the areas found to be watermarked. The models are jointly trained at low resolution and without perceptual constraints, then post-trained for imperceptibility and multiple watermarks. Experiments show that WAM is competitive with state-of-the art methods in terms of imperceptibility and robustness, especially against inpainting and splicing, even on high-resolution images. Moreover, it offers new capabilities: WAM can locate watermarked areas in spliced images and extract distinct 32-bit messages with less than 1 bit error from multiple small regions – no larger than 10% of the image surface – even for small $256 \times 256$ images. Training and inference code and model weights are available at github.com/facebookresearch/watermark-anything.

## 1 Introduction

Invisible image watermarking embeds information into image pixels in a way that is imperceptible to the human eye and yet robust. It was initially developed for intellectual property and copy protection, such as by Hollywood studios for DVDs. However, the applications of watermarking are evolving, particularly in light of the recent development of generative AI models (Kušen & Strembeck, 2018). Regulatory acts such as the White House executive order (USA, 2023), the Californian bill, the EU AI Act (Parliament & Council, 2024), and Chinese AI governance rules (of the People's Republic of China, 2023) require AI-generated content to be easily identifiable. They all cite watermarking as either compulsory or a recommended measure to detect and label AI-generated images.

Image splicing is one of the most common manipulations, whether applied for benign or malicious purposes (Christlein et al., 2012; Tralic et al., 2013). Splicing involves adding text or memes on a large portion of the image or extracting parts of images and overlaying them on others (Douze et al., 2021). It can bypass the state-of-the-art watermarking techniques, which take one global decision per image under scrutiny. Indeed, in traditional watermarking, the watermark signal fades away and is no longer detected as the surface of the watermarked area decreases. Besides, these techniques poorly answer the paradoxical question of deciding whether an image should be considered watermarked if only a small part carries the watermark. A positive decision triggered by a small area might be unfair to artists who use AI models for inpainting or outpainting. On the other hand, not being robust enough to splicing opens the door to easy removal.

To address these issues, this paper redefines watermarking as a segmentation task, giving birth to the Watermark Anything Models (WAM). Our motivation is to disentangle the strength of the watermark signal from its pixel surface, in contrast to traditional watermarking. More precisely, the WAM extractor detects if the watermark is present and extracts a binary string *for every pixel* rather than predicting a message for the whole image. These outputs are post-processed according to the final task. For global detection, the image is deemed watermarked if the proportion of watermarked pixels exceeds a user-defined threshold. For global decoding, a majority vote recovers the hidden message. A new application, out of the reach of traditional robust watermarking, is the localization of watermarked areas and the extraction of multiple hidden messages. For that purpose, we choose to apply the DBSCAN clustering algorithm over the pixel-level binary strings because it does not require any prior on the number of watermarks (or centroids). This is detailed in Sec. 3.

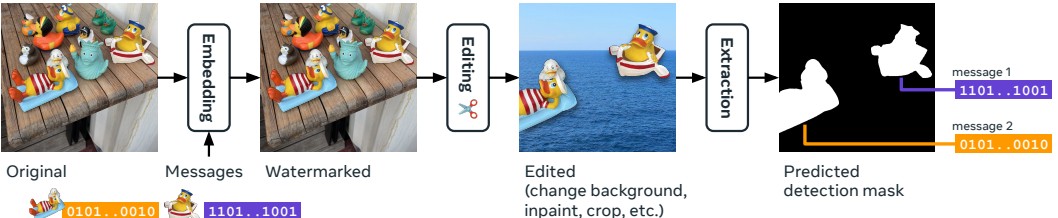

Figure 1: Overview. (a) The embedder creates an imperceptible image modification. (b) Traditional transformations (cropping, JPEG compression, etc.) and/or advanced manipulations (mixing watermarked and non-watermarked images, inpainting, etc.) may be applied to the image. (c) The extraction creates a segmentation map of watermarked parts and retrieves one or several messages.

These new functionalities require a training with new objectives, that is split into two phases. The first phase pre-trains the embedder and extractor models for low-resolution images. It essentially targets the robustness criterion. The *embedder* encodes a $n_{\text{bits}}$-bit message into a watermark signal that is added to the original image. The *augmenter* randomly masks the watermark in parts of the image and augments the result with common processing techniques (e.g., cropping, resizing, compression). The *extractor* then outputs a $(1 + n_{\text{bits}})$-dimensional vector per pixel to predict the parts of the image that are watermarked and decode the corresponding messages. Detection and decoding losses are used as training objectives. The second training phase targets the following new objectives: (1) minimize the watermark's visibility in alignment with the human visual system, (2) allow for multiple messages within the same image. This two-stage training is less prone to instability, compared to previous use of adversarial networks and divergent objectives (Zhu et al., 2018). It also trains the extractor on both watermarked and non-watermarked images, for the first time in the literature. This increases the performance and the robustness of the detection.

We first compare WAM with state-of-the-art methods for regular tasks of watermark detection and decoding on low and high-resolution images. Our results show that WAM achieves competitive performance in terms of imperceptibility and robustness. To further highlight the advantages of WAM, we then evaluate its performance on tasks that are not considered in the literature. Namely, we evaluate the localization accuracy between the predicted watermarked areas and the original mask and assess the ability to detect and decode multiple watermarks in a single image. For instance, when hiding five 32-bit messages, each in a 10% area of the image, detection of watermarked areas achieves more than 85% mIoU, even after images are horizontally flipped and the contrast adjusted, and bit accuracy (for a total of 160 bits) achieves more than 95% under the same augmentation (Sec. 5.5).

In summary, our contributions are:

- the definition of watermarking as a segmentation task;
- a two-stage training able to strike a good trade-off between invisibility and robustness even for multiple watermarks and high-resolution images;
- WAM, an embedder/extractor model competitive with state-of-the-art methods;
- the highlight of new capabilities, localization of watermarks and extraction of multiple messages as depicted in Fig. 1, together with specially designed evaluations.

## 2 RELATED WORK

**Semantic segmentation** aims to predict a category label for every pixel in an image. FCN (Long et al., 2015) employs a convolutional network predicting pixel-wise classification logits. More recent works (Zheng et al., 2021; Strudel et al., 2021) aggregate hierarchical context and are based on a ViT encoder (Dosovitskiy, 2020), followed by a decoder that generates the pixel-level predictions – similar to our extractor's architecture. On the other hand, instance segmentation identifies individual objects within an image (He et al., 2017; Carion et al., 2020). Recent literature, such as MaskFormer (Cheng et al., 2021) and Mask2former (Cheng et al., 2022), unifies semantic and instance segmentation. Segment Anything (Kirillov et al., 2023) and follow-ups (Zhang et al., 2023; Zhao et al., 2023a; Ke et al., 2024; Ma et al., 2024; Ravi et al., 2024) go one step further. They now allow users to segment any object in images or videos using prompts, which are points, bounding boxes, or natural language.

**Robust watermarking** was first designed for copyright protection (Cox et al., 2007). Traditional methods operate either in the spatial domain (Van Schyndel et al., 1994; Nikolaidis & Pitas, 1998) or in the frequency domain modifying components through an invertible transform (e.g., Fourier, Wavelet) (Cox et al., 1997; Urvoy et al., 2014). These methods were progressively replaced by deep-learning ones which remove the expert need to crafting the transforms such that the embedding is imperceptible and robust. The first approach (Vukotić et al., 2018; 2020; Fernandez et al., 2022; Kishore et al., 2022; Chen et al., 2022) embeds the watermark into the representations of an off-the-shelf pre-trained model. Our method is inspired by the second approach, which jointly trains deep learning-based architectures to embed and extract watermarks while being robust to augmentations seen during training, such as HiDDeN (Zhu et al., 2018) and followers (Zhang et al., 2019b; Luo et al., 2020; Tancik et al., 2020; Ma et al., 2022; Bui et al., 2023b;a; Pan et al., 2024).

The two tasks of watermarking, detection and decoding, are rarely performed together. Detection of the watermark, a.k.a., zero-bit watermarking, distinguishes watermarked content from original images. Decoding of a hidden message, a.k.a., multi-bit watermarking, implicitly assumes that the content under scrutiny is watermarked. One possibility for combining the two tasks is to reserve $n_{det}$ bits of the message as a detection segment – that should match a fixed pattern – with the remaining bits carrying the actual payload. Assuming that the message decoded from a non-watermarked image is random, the False Positive Rate equals $2^{-n_{det}}$. However, a recent study (Fernandez et al., 2023a, App. B.5) shows that the decoded bits are neither equiprobable nor independent. In other words, it is difficult to decode a hidden message while being confident in detecting a watermark.

Very few papers in the literature deal with watermarking objects in pictures (Bas et al., 2001; Barni et al., 2005). This trend was abandoned together with the object-oriented MPEG-4 video codec.

**Active tamper localization** relies on *semi-fragile* watermarking. Areas where the watermark is not recovered are deemed tampered. This idea appears early in the literature (Kundur & Hatzinakos, 1999; Lin et al., 2000), although deep-learning methods offer significant improvements (Asnani et al., 2022). There is a trade-off between the granularity of the localization of the tampered areas and the semi-fragility of the watermark. The biggest difficulty is to design a watermark robust to benign transformations (e.g., image compression) but fragile to malicious editing.

Our method combines detection and decoding together with a precise segmentation of the watermarked areas. This natively answers applications ranging from copyright protection, detection of AI-generated objects in images and tampering localization. EditGuard (Zhang et al., 2024) sequentially embeds a fragile watermark for localization and a robust watermark for copyright protection. Although the approach shares similarities with ours, it is not robust to geometric augmentations such as cropping or perspective changes. In contrast, our approach is conceptually simpler since a single embedding is used for detection and decoding. It offers better robustness, and enables the extraction of multiple watermarks. The approach most similar to ours is AudioSeal (San Roman et al., 2024), which introduces localized watermarking in the audio domain, but does not handle multiple messages.

An extended related work with more details on the methods mentioned above is presented in App. B.

## 3 DETECTION, LOCALIZATION, AND MESSAGE EXTRACTION

Before introducing our method and its training, this section presents several applications, i.e., how to use WAM's extractor outputs for watermark localization, zero-bit detection, and decoding of multiple messages within the same image.

The key feature is the extractor $\text{ext}_{\theta^*}$ ($\theta^*$ refers to the weights of the model after training) which outputs a tensor $y = \text{ext}_{\theta^*}(x)$ of dimensions $(1 + n_{bits}, h, w)$ for an input image $x \in \mathbb{R}^{3 \times h \times w}$. This tensor consists of a watermark detection mask $y^{det} \in [0, 1]^{1 \times h \times w}$ and a decoding mask $y^{dec} \in [0, 1]^{n_{bits} \times h \times w}$. We denote by $y_i^{det} \in [0, 1]$ the predicted detection score for pixel $i$, and by $y_i^{dec} = [y_{i,1}^{dec} \dots y_{i,n_{bits}}^{dec}] \in [0, 1]^{n_{bits}}$ its decoded message.

**Localization** (or pixel-level detection) spots watermarked pixels of the image. A pixel is deemed watermarked if its detection score $y_i^{det}$ exceeds a threshold $\tau$. Typically, we set $\tau$ empirically, by measuring the False Positive Rate (FPR) on all the pixels of a held-out training set (the FPR is the probability of a non-watermarked pixel being flagged as such).

**Detection** (or image-level detection) decides if an image is globally watermarked. From the pixel-level detection score, an image soft detection score can naturally be computed as:

$$s_{\text{det}} := \frac{1}{h \times w} \sum_{i=1}^{h \times w} \mathbb{1}\{y_i^{\text{det}} > \tau\} \in [0, 1]. \tag{1}$$

The image is flagged if $s_{\text{det}}$ is higher than a threshold that is the proportion of watermarked pixels that the user considers enough to deem a content as watermarked.

**Decoding** of a single message within an image is done by the following weighted average of the pixel-level soft predictions of the hidden message:

$$\hat{m}_k = \begin{cases} 1 & \text{if } \left[ \frac{1}{\sum_i \mathbb{1}\{y_i^{\text{det}} > \tau\}} \left( \sum_{i=1}^{h \times w} \mathbb{1}\{y_i^{\text{det}} > \tau\} \cdot y_{i,k}^{\text{dec}} \right) \right] > 0.5, \\ 0 & \text{otherwise}. \end{cases} \tag{2}$$

Decoding of multiple watermarks within one image uses a hard detection approach instead of the previous soft weighting. We isolate pixels detected as watermarked, i.e., pixels $i$ with $y_i^{\text{det}} > \tau$, and we compute the local decoded message $\tilde{m}_i$ such that for any $k \in \{1, \ldots, n_{\text{bits}}\}$, $\tilde{m}_{i,k} = \mathbb{1}\{y_{i,k}^{\text{dec}} > 0.5\}$. The DBSCAN (Density-Based Spatial Clustering of Applications with Noise) algorithm (Ester et al., 1996; Schubert et al., 2017) clusters the set of locally decoded messages. It outputs some centroids and an assignment for every pixel detected as watermarked. DBSCAN selects these centroids among the initial set, thus ensuring that the final decoded messages are binary words.

DBSCAN offers the advantage of not requiring a pre-defined number of clusters. Instead, we need to specify two parameters: (1) $\min_{\text{samples}}$, the minimum number of points for a cluster to be considered valid, and (2) $\varepsilon$, the maximum distance between two samples for one to be considered as in the neighborhood of the other. For further details on the DBSCAN algorithm, see App. C.1.

Handling multiple messages in a single image makes WAM robust against attacks that involve splicing several watermarked images together, which can compromise the decoding of traditional watermarking schemes in real-world scenarios. It also enables active object detection (Asnani et al., 2024), where the objective is to track watermarked objects. Additionally, it allows for the identification of the use of multiple watermarked AI tools.

## 4 WATERMARK ANYTHING MODELS

### 4.1 THE MODEL

WAM considers two joint models: a watermark embedder $\text{emb}_\theta$ and a watermark extractor $\text{ext}_\theta$, where $\theta$ gathers the parameters of these functions. The embedder defines the watermark procedure for hiding a message into an image, while the extractor detects watermarking and decodes messages. This section discusses our choices for the different pieces that constitute WAM. App. D.1 gives the technical details of the network architectures.

**The watermark embedder** ($\text{emb}_\theta$) consists of an encoder, a binary message lookup table, and a decoder. The encoder represents an image in a latent space. The lookup table is used to translate the message into a tensor which is concatenated to the image representation. From that, the decoder outputs a watermark signal, which is added to the original image with adequate scaling.

The autoencoder that we consider is based on the architecture of the variational autoencoder of LDM (Rombach et al., 2022). Its encoder, $\text{enc}_\theta$, compresses a $h \times w$ input image $x$ into a latent $z \in \mathbb{R}^{d_z \times h' \times w'}$ (with downsampling factor $f = h/h' = w/w'$). The binary message lookup table $\mathcal{T}_\theta$ is of shape $(n_{\text{bits}}, 2, d_{\text{msg}})$. Each bit of the message $m$ is mapped to the embedding $\mathcal{T}_\theta(k, m_k, \cdot) \in \mathbb{R}^{d_{\text{msg}}}$, depending on its position $k \in \{1, \ldots, n_{\text{bits}}\}$ and its value $m_k \in \{0, 1\}$. The $n_{\text{bits}}$ embeddings are averaged, resulting in a vector of size $d_{\text{msg}}$ which is repeated to form a tensor of shape $(d_{\text{msg}}, h', w')$. The concatenation to the image representation yields an activation of shape $(d_z + d_{\text{msg}}) \times h' \times w'$. The decoder, $\text{dec}_\theta$, maps back this activation to the watermark signal $\delta_\theta(x, m)$ of the same shape as the original image. The range of this signal is $[-1, 1]$ because the last layer of $\text{dec}_\theta$ is a hyperbolic

tangent activation. It is finally added to the original image to produce the watermarked output $x_{\mathrm{m}} = \mathrm{emb}_\theta(x) = x + \alpha \cdot \delta_\theta(x, m)$. Parameter $\alpha \in \mathbb{R}_+$ is called the watermark strength as it controls the distortion applied to the original image.

**The watermark extractor** ($\mathrm{ext}_\theta$) takes on the dual role of detecting watermarked pixels and decoding their embedded messages. We use an architecture similar to SETR (Zheng et al., 2021) and Segment Anything (Kirillov et al., 2023). It comprises a ViT encoder paired with a pixel decoder that upsamples the embeddings to the original image size. The watermark extractor outputs a vector of size $1 + n_{\mathrm{bits}}$ for every pixel, as described in Sec. 3.

**High-resolution.** WAM operates at a fixed resolution of $h \times w$. To extend it to higher resolutions, an anisotropic scaling resizes the image to $h \times w$ and $\mathrm{emb}_\theta$ computes the watermark signal $\delta$ from this resized image. A bilinear interpolation scales $\delta$ back to the size of the original image. The extraction also operates at $h \times w$, by resizing all images before feeding them to the network. Therefore the extraction process remains consistent with the pre-training conditions. We only use these interpolations at embedding time. WAM is thus only trained on low resolution images (Sec. 4.2) but can be used for high-resolution images too, which represents an important training compute gain. A similar approach also appears in the recent literature (Bui et al., 2023a).

## 4.2 Pre-training models for localized message embedding and extraction

The objective of this first training phase is to obtain a robust and localizable watermark hiding a $n_{\mathrm{bits}}$-bit message inside parts of an image, without caring much about imperceptibility. This stage does not incorporate any perceptual loss: our goal is to achieve perfect localization and decoding even after severe augmentations. Figure 2 illustrates the detailed process.

**Augmentations** are two-step processes. They first splice images $x$ and $x_{\mathrm{m}}$ based on a binary mask and then apply usual image transformations to improve robustness.

The first step randomly samples a mask among full masks, rectangles, irregular shapes, or object segmentation maps (possibly provided by the training dataset). These masks are inverted with probability 0.5. It yields a mask $r_{\mathrm{mask}} \in [0, 1]^{h \times w}$. Figure 2 shows examples of masks, with more samples displayed in Figure 6 of App. D.2. The first step then computes the spliced image as $x_{\mathrm{masked}} = r_{\mathrm{mask}} \odot x_{\mathrm{m}} + (1 - r_{\mathrm{mask}}) \odot x$, where $\odot$ denotes component-wise multiplication.

The second step of the augmentation applies a processing typical from real-world image editing including geometric transformations (identity, resize, crop, rotate, horizontal flip, perspective) and valuemetric adjustments (JPEG, Gaussian blur, median filter, brightness, contrast, saturation, hue). The geometric transformation alters the positions of watermarked pixels, so it is also applied to the mask to keep track of the watermarked areas. This adjusted mask is the ground truth for the localization and it is denoted $y^{\mathrm{det}, \star}$ in the sequel. These augmentations are relatively strong to reflect

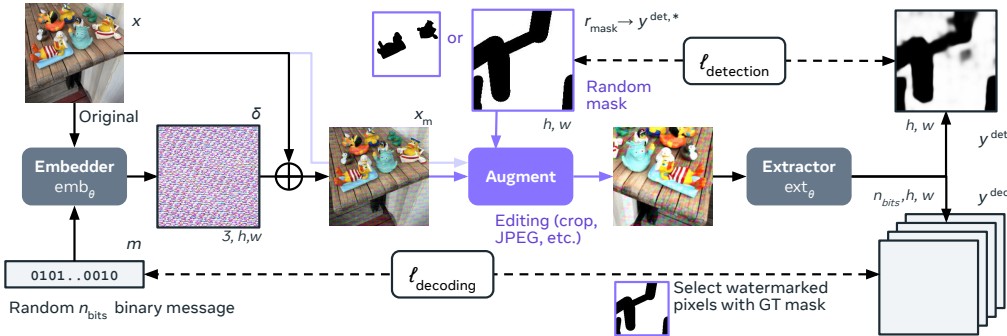

Figure 2: The first training phase of WAM, as described in Sec. 4.2, jointly trains the *watermark embedder* to predict an additive watermark and the *watermark extractor* to detect watermarked pixels and decode the hidden message. In between, the augmenter 1) splices the watermarked and the original images based on a random mask and 2) applies classical image transformations.

the transformations found on media sharing websites, as opposed to professional photo publications. Additional details and discussions on augmentations are postponed to App. D.4.

When a given image goes through the embedder-augmentation-extractor sequence, the final output depends on the models' parameters $\theta$ (omitted so far for clarity):

$$y(\theta) = [y^{\text{det}}(\theta), y^{\text{dec}}(\theta)] = \text{ext}_\theta \Big( \text{transformation} \big( \text{masking} \big( \text{emb}_\theta(x, m), x \big) \big) \Big). \quad (3)$$

**Objectives.** The training minimizes the objective function $\ell(\theta)$ which is a linear combination of the detection and decoding losses: $\ell(\theta) = \lambda_{\text{det}} \cdot \ell_{\text{det}}(\theta) + \lambda_{\text{dec}} \cdot \ell_{\text{dec}}(\theta)$.

The detection loss is the average of the pixel-wise cross-entropy between $y^{\text{det}}(\theta) \in [0, 1]^{h \times w}$ and the ground truth $y^{\text{det},\star} \in \{0, 1\}^{h \times w}$ (pixel watermarked or not). Similarly, the decoding loss is the average of the pixel-wise and bit-wise binary cross-entropy between $y_{i,k}^{\text{dec}}(\theta)$ and $m_k$ only over the watermarked pixels, where $m$ is a random message originally embedded in that image. For a given image and message, $\ell_{\text{det}}$ and $\ell_{\text{dec}}$ are:

$$\ell_{\text{det}}(\theta) = \frac{-1}{h \times w} \sum_{i=1}^{h \times w} \left[ y_i^{\text{det},\star} \log(y_i^{\text{det}}(\theta)) + (1 - y_i^{\text{det},\star}) \log(1 - y_i^{\text{det}}(\theta)) \right], \quad (4)$$

$$\ell_{\text{dec}}(\theta) = \frac{-1}{n_{\text{bits}} \times \sum_{i=1}^{h \times w} y_i^{\text{det},\star}} \sum_{i=1}^{h \times w} y_i^{\text{det},\star} \sum_{k=1}^{n_{\text{bits}}} \left[ m_k \log(y_{i,k}^{\text{dec}}(\theta)) + (1 - m_k) \log(1 - y_{i,k}^{\text{dec}}(\theta)) \right]. \quad (5)$$

### 4.3 POST-TRAINING FOR IMPERCEPTIBILITY AND MULTIPLE WATERMARKS

The model trained in the first phase (Sec. 4.2) produces a too visible watermark. Moreover, it cannot deal within multiple watermarks in an image. The second training phase addresses these issues.

**Perceptual heatmap.** The Just-Noticeable-Difference (JND) map is a hand-crafted model of the minimum artifact perceivable by a human at every pixel. For instance, artifacts on flat, uniform areas are more visible than on textured ones. The JND map was introduced by Chou & Li (1995) and later used by Wu et al. (2017). App. C.2 gives more details on its computation.

Given an image $x$, we have $\text{JND}(x) \in \mathbb{R}^{3 \times h \times w}$. We modulate the intensity of the watermark per pixel at embedding time: the final watermarked image is computed as $x_{\text{m}} = x + \alpha_{\text{JND}} \cdot \text{JND}(x) \odot \delta_\theta(x, m)$. In practice, we found that applying this JND on the model trained in the first phase slightly degrades the detection and decoding performance. The fine-tuning cancels this degradation. Applying JND only in the second training phase makes training easier than previous methods relying on "perceptual" or "contradictory" losses (like StegaStamp (Tancik et al., 2020) or HiDDeN (Zhu et al., 2018)).

**Multiple watermarks.** Since the first training phase (Sec. 4.2) uses a single message, it tends to decode a constant message across pixels even on image regions with different messages. The second training phase introduces several masks to address this limitation. The masks are generated as before with distinct random messages in each of the masked areas. The number of disjoint masks goes from 1 to 3 with probabilities 0.6, 0.2, and 0.2 respectively (see App. D.2).

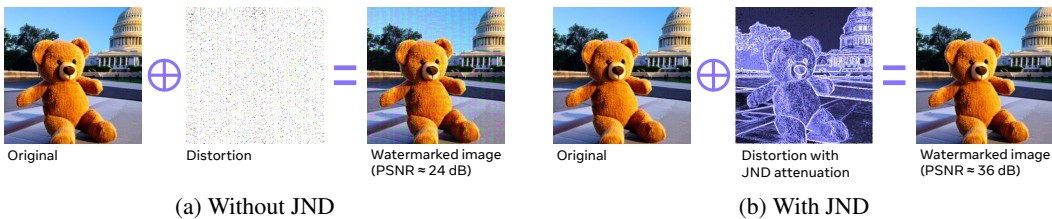

(a) Without JND                                    (b) With JND

Figure 3: Impact of the JND map on imperceptibility. (Left) After the first training phase, the watermark is highly perceptible. (Right) When applying the JND attenuation, it is hidden in areas where the eye is not sensitive to changes, which makes it less visible. Fine-tuning with the JND recovers the initial robustness. The difference is displayed as $10 \times \text{abs}(x_{\text{m}} - x)$.

Table 1: Evaluation of the watermark imperceptibility. We report the PSNR, SSIM, and LPIPS between watermarked and original images of COCO (low/mid-resolution) and DIV2k (high-resolution).

| | | HiDDeN | DCTDWT | SSL | FNNS | TrustMark | WAM (ours) |
|---|---|---|---|---|---|---|---|
| COCO | PSNR (↑) | 38.2 | 37.0 | 37.8 | 37.7 | 40.3 | 38.3 |
| | SSIM (↑) | 0.98 | 0.98 | 0.98 | 0.98 | 0.99 | 0.99 |
| | LPIPS (↓) | 0.05 | 0.02 | 0.07 | 0.06 | 0.01 | 0.04 |
| DIV2K | PSNR (↑) | 38.4 | 38.7 | 38.2 | 39.0 | 39.1 | 38.8 |
| | SSIM (↑) | 0.98 | 0.99 | 0.98 | 0.99 | 0.99 | 0.99 |
| | LPIPS (↓) | 0.07 | 0.03 | 0.11 | 0.04 | 0.01 | 0.03 |

The detection loss $\ell_{\text{det}}$ takes the union of all masks as ground truth $y_i^{\text{det},\star}$. The decoding loss $\ell_{\text{dec}}$ is computed separately for each message and the losses are summed up. The scenario of multiple watermarks within a single image arises when watermarked images are combined. Unlike other methods producing a global message, WAM can now distinguish and decode each message separately.

# 5 EXPERIMENTS & RESULTS

## 5.1 IMPLEMENTATION DETAILS

We provide here further implementation details using the notations introduced in Sec. 4.1, and the rest of the architectures are described in App. D.1. All experiments are run with $n_{\text{bits}} = 32$. There is a total of 1.1M parameters for the embedder and 96M for the extractor (equivalent to a ViT-base). In our experiments, we found it necessary to increase the size of the extractor compared to the embedder. This is probably because the extractor has more tasks to handle, such as segmenting and extracting multiple messages, while the embedder only needs to embed one message into an image. Also, it is important for the embedding process to be quick since it happens on the user-side, so we keep it voluntarily small to be fast and usable at scale. We train our model on the MS-COCO training set with blurred faces (Lin et al., 2014), that contains 118,000 images, many of them with segmentation masks. We train at resolution $h \times w = 256 \times 256$, and with $f = 8$ (so $h' \times w' = 32 \times 32$). The first training phase (Sec. 4.2) is optimized with AdamW (Kingma, 2014; Loshchilov, 2017) with a linear warmup of the learning rate in 5 epochs from $1 \times 10^{-6}$ to $1 \times 10^{-4}$ and a cosine annealing to $1 \times 10^{-6}$. We set $\lambda_{\text{dec}} = 10$, $\lambda_{\text{det}} = 1$, $\alpha = 0.3$. We train with a batch size of 16 per GPU for 300 epochs using 8 V100 GPUs which takes roughly 2 days. The second training phase (Sec. 4.3) further trains the model with the JND attenuation for 200 epochs, hiding up to 3 messages per image using either randomly sampled rectangles or segmentation masks. During this phase, $\alpha_{\text{JND}} = 2$. App. D.2 provides details on the generation of masks used during both training phases. For the extraction of multiple messages, we use the Scikit-learn DBSCAN implementation (Pedregosa et al., 2011).

## 5.2 QUALITY

Table 1 evaluates quantitatively the difference between the watermarked and original images with PSNR, SSIM and LPIPS (Zhang et al., 2018) metrics. We adapt the methods so that they are comparable in terms of imperceptibility. We show qualitative examples for COCO and DIV2k images in App. F. The perceptual quality is good: the JND map successfully modulates the watermark signal in areas where the eye is not sensitive. Nevertheless, repetitive and regular patterns can be observed in some images when the bright or textured areas are big enough (furs of animals for instance).

## 5.3 DETECTION AND DECODING

We benchmark WAM's performance against several watermarking methods: DCTDWT (Al-Haj, 2007), HiDDeN (Zhu et al., 2018), TrustMark (Bui et al., 2023a), SSL (Fernandez et al., 2022), FNNS (Kishore et al., 2022). We also compare WAM to generation-time watermarking methods (Fernandez et al., 2023a; Wen et al., 2023) in App. E.1. Although this list is not exhaustive, it covers the main types of state-of-the-art methods. For HiDDeN, we use a model which hides 48 bits. FNNS,

SSL, and DCTDWT can hide messages or arbitrary length, we choose to hide 48 bits as well. We use the first 16 bits for detection and 32 bits for message decoding. For detection, an image is flagged as watermarked if at most 1 bit is wrongly decoded. This corresponds to a theoretical False Positive Rate (FPR) of $2.6 \times 10^{-4}$, i.e., 2.6 out of 10,000 images are falsely flagged as watermarked on average. TrustMark (Bui et al., 2023a) directly outputs a boolean (watermarked or not). We use the version that hides 40 bits, but only use the first 32 bits. For all evaluations, we apply the watermark embedding and extraction at the original image resolution as described in the high-resolution paragraph of Sec. 4.1. Finally, for WAM, an image is flagged if $s_{det}$ (Eq. 1) is higher than 0.07. This value empirically delivers an approximately similar FPR on the COCO validation set.

We evaluate the robustness against various transformations: *geometric* (flip, crop, perspective, and rotation), *valuemetric* (adjustments in brightness, hue, contrast or saturation, median or Gaussian filtering, and JPEG compression), *splicing*: scenarios where only 10% of the image is watermarked superimposed onto the original or an other background. We also evaluate the robustness against *inpainting*, using the LaMa (Suvorov et al., 2022) model to inpaint areas of the watermarked image specified by random or segmentation masks. For COCO, we use the union of all segmentation masks for each image which in average corresponds to 30% of the image, and for DIV2k we use a randomly generated mask (because there are no segmentation masks) that covers around 35% of the image. We give details on the compared baselines in App. D.3 and on the transformations in App. D.4.

Table 2a presents the detection and decoding results for low/mid-resolution images, averaged over the first 10k images of the COCO validation set – the detailed results for every augmentation used to form the different groups are in App. E.3. Table 2b does the same for high-resolution images from the DIV2k (Timofte et al., 2018) validation set. For both distributions, WAM is competitive and even shows improved robustness on classical *geometric*/*valuemetric* transformations. Most importantly, WAM performs better on *splicing* or *inpainting* where other methods fail to achieve more than 90% TPR and Bit acc. This is true even if WAM was not explicitly trained on high resolution images or to be robust against inpainting. We also show examples of WAM's detection masks after inpainting in Fig. 10 of App. E, where some modified parts of images are not detected as watermarked anymore.

## 5.4 LOCALIZATION

WAM and EditGuard (Zhang et al., 2024) are the only methods that provide watermark localization. We therefore consider images at fixed resolution $512 \times 512$ (unlike in Sec. 5.3) to align with the setup of EditGuard. We nevertheless observe similar results at different resolutions for WAM. We focus on the COCO validation set and splice centered watermarked rectangles of various sizes within the images to ensure a well-controlled experiment. We then either apply no transformation or crop the upper left part of the image (25%), which is then resized to the original size. By doing so, the proportion of watermarked pixels is still the same as in the spliced image before cropping, which allows us to evaluate the robustness of the localization. Figure 7 of App. D.5 illustrates this protocol.

Figure 4 reports the mean Intersection over Union (mIoU) to evaluate the localization, as commonly done in the segmentation literature. Figure 4 evaluates the bit accuracy – through localization – following Eq. 2. We observe that WAM accurately predicts both classes, even after cropping and resizing, except when the watermarked area covers 95% of the image, in which case the extractor tends to classify all pixels as watermarked. In terms of bit accuracy, WAM recovers in average 31 out of 32 bits even when only 10% of the $256 \times 256$ image is watermarked, and around 25 bits when only 10% of a 25% crop is watermarked (which corresponds to 2.5% of the overall number of pixels). For both evaluations, WAM outperforms EditGuard which, in particular, is not robust to cropping.

## 5.5 MULTIPLE WATERMARKS

We compare the detection and decoding of multiple watermarks before and after the second training phase of WAM (Sec. 4.3). We embed up to five distinct messages into five separate 10% areas of every image (first resized to 256 to ease the experiments). This is done by feeding the image several times to WAM's embedder, then pasting the different watermarked areas onto the original image. These areas are squares disposed in a checkerboard pattern, which ensures that the watermarked areas do not overlap and have the same size (see Fig. 8 of App. D.6). This is an arbitrary choice made to remove confounding factors during evaluation, although the training does not require the watermarked areas to be squared nor to have the same size. Following the methodology detailed in Sec. 3, we apply the

Table 2: Detection and decoding after image editing (detailed in Sec. D.4). We show the bit accuracy (Bit acc.) between the encoded and decoded messages, the proportion of images correctly deemed watermarked (TPR), and the proportion of non-watermarked images falsely detected as watermarked (FPR), in %. Since HiDDeN, DCTDWT, SSL and FNNS do not naturally provide a detection result, we hide 48 bits and reserve 16 bits for detection, and flag an image as watermarked if it has strictly less than two bits incorrectly decoded (these baselines are detailed in Sec. 5.3).

(a) On the first 10k validation images of COCO (low to mid resolution)

| | | Augmentations | | | | | | | | | |
|---|---|---|---|---|---|---|---|---|---|---|---|
| | | None | | Geometric | | Valuemetric | | Inpainting | | Splicing | |
| Method | FPR | TPR | Bit acc. | TPR | Bit acc. | TPR | Bit acc. | TPR | Bit acc. | TPR | Bit acc. |
| HiDDeN | 0.08 | 76.9 | 95.5 | 31.2 | 80.1 | 48.4 | 87.2 | 44.7 | 88.7 | 0.9 | 68.0 |
| DWTDCT | 0.02 | 77.1 | 91.4 | 0.0 | 50.5 | 13.6 | 58.1 | 47.8 | 81.7 | 0.8 | 59.9 |
| SSL | 0.00 | 99.9 | 100.0 | 14.3 | 76.5 | 70.5 | 92.1 | 67.7 | 91.1 | 0.4 | 58.9 |
| FNNS | 0.10 | 99.6 | 99.9 | 62.4 | 86.6 | 82.1 | 93.9 | 89.4 | 97.3 | 38.7 | 88.5 |
| TrustMark | 2.88 | 99.8 | 99.9 | 36.3 | 71.4 | 90.1 | 98.2 | 39.4 | 83.2 | 83.2 | 57.1 |
| WAM (ours) | 0.04 | 100.0 | 100.0 | 99.3 | 91.8 | 100.0 | 99.9 | 97.9 | 99.2 | 100.0 | 95.3 |

(b) On the 100 validation images of DIV2k (high resolution)

| | | Augmentations | | | | | | | | | |
|---|---|---|---|---|---|---|---|---|---|---|---|
| | | None | | Geometric | | Valuemetric | | Inpainting | | Splicing | |
| Method | FP | TPR | Bit acc. | TPR | Bit acc. | TPR | Bit acc. | TPR | Bit acc. | TPR | Bit acc. |
| HiDDeN | 0 | 72.0 | 95.8 | 33.5 | 81.3 | 53.3 | 88.1 | 53.3 | 88.1 | 1.5 | 70.2 |
| DWTDCT | 0 | 75.0 | 88.9 | 0.0 | 50.6 | 18.4 | 58.9 | 73.0 | 86.7 | 31.5 | 71.6 |
| SSL | 0 | 100.0 | 100.0 | 32.5 | 83.6 | 75.8 | 92.8 | 95.0 | 96.1 | 0.5 | 59.8 |
| FNNS | 0 | 97.0 | 99.9 | 63.5 | 87.2 | 79.8 | 93.9 | 94.0 | 99.2 | 48.5 | 89.5 |
| TrustMark | 5 | 100.0 | 100.0 | 33.1 | 70.5 | 87.4 | 97.9 | 0.0 | 72.1 | 1.5 | 57.3 |
| WAM (ours) | 0 | 100.0 | 99.9 | 96.1 | 89.0 | 100.0 | 99.9 | 100.0 | 99.8 | 99.5 | 94.2 |

DBSCAN algorithm to the vector outputs $\tilde{m}_i$ of the extractor corresponding to all pixels $i$ identified as watermarked ($y_i^{\text{det}} > \tau$). It identifies clusters corresponding to different watermarked regions without requiring the number of hidden messages to be known in advance. We use $\tau = 0.5$ to threshold the watermarked pixels, and we choose a rather strict setup with $\varepsilon = 1$ and $\min_{\text{samples}} = 1000$ pixels (around 2% of the image) in our evaluations, resulting from an hyperparameter search detailed in Fig. 9 of App. E.2. This means that a cluster will be considered only if it has at least 2% of the image's pixels, and that the maximum distance between two predicted messages to be neighbors is 1.

Figure 5 presents the results. We compute the bit accuracies by comparing the centroid of each detected cluster (decoded message) with the ground truth message that has the largest overlapping area: the bit accuracy is thus computed only across the clusters that are discovered by DBSCAN.

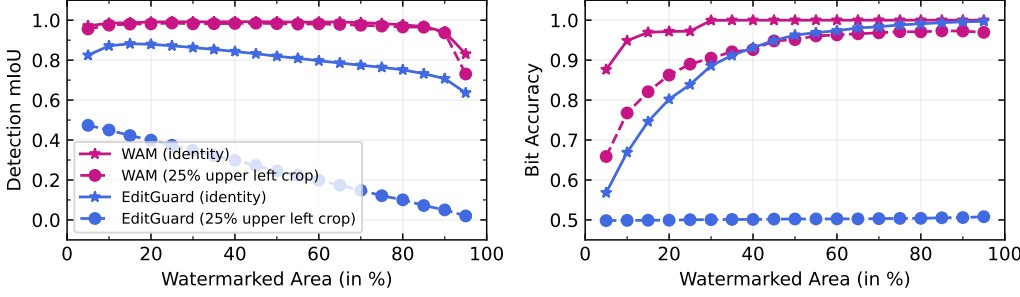

Figure 4: Evaluation of the localization on the validation set of COCO, with or without cropping before extraction, following the setup described in Sec. 5.4. (Left) Localization accuracy using intersection over union between the predicted watermarked areas and the ground-truth mask. (Right) Bit accuracy between the ground truth message and the decoded message, computed from Eq. (2).

Without the second phase of training, the messages get mixed up, and WAM predicts the same (wrong) message for all watermarked pixels. After the training, WAM is able to accurately extract up to five different 32-bit messages from the areas covering 10% of the image each. Therefore, although the second phase of training embeds between 1 and 3 watermarks per image, WAM generalizes to more watermarks. It also shows that WAM's effective capacity is greater than 32 bits (since it can hide multiple messages) if we do not consider heavy crops as a transformation we want the method to be robust against, similarly as (Bui et al., 2023b; Tancik et al., 2020; Wen et al., 2023).

This method remains effective after some image manipulations; extended quantitative results are presented in Fig. 9 of App. E.2. For example, WAM achieves 85% mIoU even when the images have been horizontally flipped and the contrast adjusted. Furthermore, the bit accuracy, averaged across 5 messages totaling 160 bits, exceeds 95% under the same conditions. However, WAM fails when JPEG compression is also added on top of these two transformations. Illustrative examples of the identified clusters in various scenarios can be found in Fig. 8 of App. D.6.

## 6   CONCLUSION, LIMITATIONS, AND FUTURE WORK

**Conclusion.**   This work introduces the Watermark Anything Model, which approaches image watermarking as a segmentation task. WAM is able to predict whether an image is watermarked or not, as well as to localize its watermarked regions. It thus handles images with a small watermarked part, or where parts of the image have been removed or edited. Additionally, it is able to detect and extract multiple watermarks within the same image, for the first time in the literature. Our training which introduces localization under heavy augmentations also offers state-of-the-art robustness on classical settings considered by current watermark methods, where most of the image is watermarked.

We identify two main limitations that we address in the next paragraphs.

**Low payload.**   In our experiments, WAM's capacity is limited to 32 bits, and training on larger messages is challenging. In contrast, other watermarking methods such as EditGuard or RoSteALS can successfully embed more than 100 bits per image, but are not robust to crops or outpainting. This is a trade-off between capacity and robustness. Note that in practice, other methods use decoding to match with the original message and conclude if an image is watermarked. In contrast, WAM's detection process is separate from the decoding, so 32 bits may be enough for most applications.

**Perceptual quality.**   In spite of the JND weighting, we notice that the watermark can still be visible in some areas of the watermarked images, even at a relatively high PSNR (see examples in App. F). This could be due to the JND map focusing only on the cover image and not on the watermark signal itself. Improvements might be achieved by employing more sophisticated Human Visual System (HVS) models or by regularizing the watermark to eliminate repetitive patterns during training.

**Post-ICLR rebuttal changes.**   We added Appendix G to further discuss WAM's operating point and include additional experiments: more baseline comparisons, robustness to diffusion-based purification, neural compression, overlapping watermarks, and balancing the perceptibility/robustness trade-off.

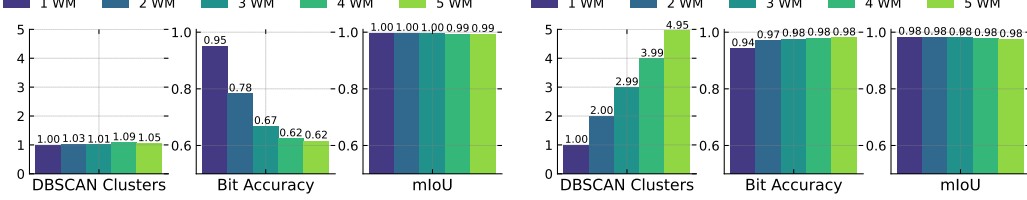

(a) After first training phase (PSNR ≈ 25 dB)          (b) After second training phase (PSNR ≈ 38 dB)

Figure 5: Results on multiple watermarks extracted from a single image. We use non overlapping 10 % rectangular masks to watermark up to 5 parts of images from COCO, with different messages, and report the average number of clusters detected by DBSCAN, bit accuracy across found messages, as well as the mIoU of watermark detection on all objects. (Left) After the first training phase, the bit accuracy strongly decreases as the number of watermarks grows. (Right) After fine-tuning, it stays roughly constant no matter the number of watermarked parts. The mIoU stays stable in both cases.

## REPRODUCIBILITY STATEMENT

Section 5 provides the specifics of our training and evaluation setup. Further implementation details, including network architectures, mask designs, transformation parameters for robustness evaluation, and settings for baseline comparisons, are presented in App. D. The models and code for inference and training used in this paper will be made available upon publication. The code is based on PyTorch, the training dataset is publicly available, and the training requires half a week on 8 V100 GPUs, which makes the models reproducible at reasonable cost.

The training and inference code, as well as trained models are available at: `https://github.com/facebookresearch/watermark-anything`

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

## A  ETHICAL STATEMENT

### A.1  SOCIETAL IMPACT

Watermarking in general improves the traceability of content, be it AI-generated, or not. It can have positive consequences, for example when it is used to trace the origin of fake news or to protect intellectual property. This traceability can also have negative consequences, for example when it is used to trace political opponents in authoritarian regimes or whistleblowers in secretive companies. Besides, it is not clear how to disclose watermark detection results, which may foster a closed ecosystem of detection tools. It may also exacerbate misinformation by placing undue emphasis on content that is either not detected, generated by unknown models, or authentic but used out of context. We however believe that the benefits of watermarking outweigh the risks, and that the development of robust watermarking methods is a positive step for our society.

### A.2  ENVIRONMENTAL IMPACT

The cost of the experiments and of model training is high, though order of magnitude less than other computer vision fields (Oquab et al., 2023). One training with a schedule similar to the one reported in the paper represents $\approx 30$ GPU-days. We also roughly estimate that the total GPU-days used for running all our experiments to $5000$, or $\approx 120$k GPU-hours. This amounts to total emissions in the order of 20 tons of $CO_2$eq. Estimations are conducted using the Machine Learning Impact calculator presented by Lacoste et al. (2019). We do not consider in this approximation: memory storage, CPU-hours, production cost of GPUs/ CPUs, etc.

We were careful to limit the environmental impact of our research by doing most of our experiments on smaller models, on fewer epochs (100) and at lower resolutions ($128 \times 128$), before scaling up to larger models, more epochs and higher resolutions. We also focus on small specialized model that are efficient at inference time and more environmental-friendly (Luccioni et al., 2024). At the end of the day, we believe that the environmental cost of this research is justified by the potential benefits of WAM.

## B  EXTENDED RELATED WORK ON IMAGE WATERMARKING

### B.1  TRADITIONAL WATERMARKING

We call traditional watermarking a technique embedding a digital watermark in a host content. As far as images are concerned, the first methods are usually classified into two categories depending on the space in which the watermark is embedded. In *spatial domain*, the watermark is encoded by directly modifying pixels, such as flipping low-order bits of selected pixels (Van Schyndel et al., 1994). For example, Nikolaidis & Pitas (1998) slightly modify the intensity of randomly selected image pixels while taking into account properties of the human visual system, robustly to JPEG compression and lowpass filtering. Bas et al. (2002) create content descriptors defined by salient points and embed the watermark by adding a pattern on triangles formed by the tessellation of these points. Ni et al. (2006) use the zero or the minimum points of the histogram of an image and slightly modifies the pixel grayscale values to embed data into the image. The second category *frequency domain* watermarking offers better robustness. It usually spreads a pseudorandom noise sequence across the entire frequency spectrum of the host signal (Cox et al., 1997). The first step is a transformation that computes the frequency coefficients. The watermark is then added to these coefficients, taking into account the human visual system. The coefficients are mapped back onto the original pixel space through the inverse transformation to generate the watermarked image. The transform domains include Discrete Fourier Transform (DFT) (Urvoy et al., 2014), Quaternion Fourier Transform (QFT) (Bas et al., 2003; Ouyang et al., 2015), Discrete Cosine Transform (DCT) (Bors & Pitas, 1996; Piva et al., 1997; Barni et al., 1998; Li et al., 2011), Discrete Wavelet Transform (DWT) (Xia et al., 1998; Barni et al., 2001; Furon & Bas, 2008), both DWT and DCT (Feng et al., 2010; Zear et al., 2018), etc.

Deep learning-based methods have recently emerged as alternatives for traditional watermarking. The first attempts use neural networks as a fixed transform into a latent space (Vukotić et al., 2018; 2020; Kishore et al., 2022; Fernandez et al., 2022) Since there is no inverse transform, the embedding is done iteratively by gradient descent over the pixels. The recent approaches are often built as

embedder/extractor networks. They are trained end-to-end to invisibly encode information while being resilient to transformations applied during training. This makes it easier to build robust systems and avoids algorithms hand-crafted for specific transformations. HiDDeN (Zhu et al., 2018) is a famous representative of this approach and has been extended in several ways. Luo et al. (2020) add adversarial training in the attack simulation to bring robustness to unknown transformations. Zhang et al. (2019b; 2020); Yu (2020) use an attention filter further improving imperceptibility. Ahmadi et al. (2020) adds a circular convolutional layer that helps spreading the watermark signal over the image. Wen & Aydore (2019) use robust optimization with worst-case attack as if an adversary were trying to remove the mark. Many other approaches focused on improving robustness, imperceptibility, speed, etc. (Jia et al., 2021; Bui et al., 2023b;a; Huang et al., 2023; Evennou et al., 2024; Pan et al., 2024).

Another line of works focuses on steganography (Baluja, 2017; Wengrowski & Dana, 2019; Zhang et al., 2019a; Tancik et al., 2020; Jing et al., 2021; Ma et al., 2022) that pursues a different goal. Steganography hides a message in the image without leaving any statistical traces, but the robustness is null or limited.

## B.2 Watermarking for generative AI

The most trendy research direction in watermarking is arguably the detection of AI-generated images for transparency and for filtering such results when building new models. The *Post-generation* or *post-hoc* approach uses traditional watermarking: the content is first generated and then watermarked. On the contrary, the *generation-time* methods natively generate watermarked images. Watermarking no longer incurs additional runtime and is more robust and secure than post-hoc. We broadly categorize generation-time watermarking into the two following categories.

*In-model* methods modify the weights of the generative model. This allows open-sourcing the model without revealing the watermark. The earliest methods (Wu et al., 2020; Yu et al., 2021; Zhao et al., 2023b) watermark the images of the training set with the hope that the model learns what watermark is during its training. This is computationally expensive and not scalable. Alternatively, some proposals (Fei et al., 2022; 2024) train Generative Adversarial Networks (GAN) with additional watermarking losses such that generated images contain the watermark. Stable Signature (Fernandez et al., 2023a) focuses on Latent Diffusion Models (LDM) and fine-tunes the latent decoder to embed the watermark, while Feng et al. (2024) fine-tune the U-Net that predicts the latent diffusion noise instead. To eliminate the need to fine-tune the model for every user, some papers use a hyper-network predicting the modifications to be applied to the generative model, be it a GAN (Yu et al., 2022; Fei et al., 2023) or diffusion-based (Kim et al., 2024).

*Out-of-model* methods alter the generation process. This is easier to implement since it does not require training or fine-tuning the model. Ci et al. (2024a) and Rezaei et al. (2024) propose an adapter to the decoder that takes the secret message as input. A different class of out-of-model methods, specific to diffusion models, embed the watermark by adding patterns to the initial noise (seed). For example, Tree-Ring (Wen et al., 2023) adds circular patterns to the initial noise and inverts the diffusion process to extract the watermark. As follow-up works, Hong et al. (2024) improve the inversion of the diffusion process, Ci et al. (2024b) extend the method to multi-bit watermarking, and Lei et al. (2024) deploy the embedder/extractor framework over the noise.

## C Algorithms details

### C.1 DBSCAN

DBSCAN (Density-Based Spatial Clustering of Applications with Noise) (Ester et al., 1996; Schubert et al., 2017) is a clustering method that groups points in a dataset based on their density and proximity to each other. In our case, points are locally decoded messages (one per pixel deemed as watermarked). It works as follows:

1. Initialization: Set the parameters $\varepsilon$ (maximum distance between two samples for one to be considered as in the neighborhood of the other) and $\min_{\text{samples}}$ (minimum number of samples in a cluster).

2. Neighborhood search: For each point $p$ in the dataset, find all points within a distance of $\varepsilon$ from $p$. This forms the neighborhood of $p$.

3. Cluster formation: If the neighborhood of $p$ contains at least $\min_{\text{samples}}$ points, form a cluster around $p$. Otherwise, mark $p$ as noise.

4. Cluster expansion: For each point $q$ in the cluster, find all points within a distance of $\varepsilon$ from $q$. Add these points to the cluster if they are not already part of it.

5. Repeat steps 3-4: Continue expanding the cluster until no more points can be added.

6. Assign labels: Assign a label to each point in the dataset indicating which cluster it belongs to (if any).

It is important to note that $\varepsilon$ is neither a hard boundary nor a maximum bound on the distances of points within a cluster: points that are further apart than $\varepsilon$ but still within the neighborhood of a core point (a point with at least $\min_{\text{samples}}$ neighbors within $\varepsilon$ distance) can also be part of the same cluster.

One of the significant advantages of DBSCAN is its ability to cluster points without knowing the number of clusters beforehand, unlike some other clustering algorithms like K-Means. Additionally, it identifies arbitrarily shaped clusters (watermarked area in our case), including those that may be surrounded by a different cluster. This clustering is also robust to outliers. However, DBSCAN also has some limitations. It is sensitive to hyper-parameters $\varepsilon$ and $\min_{\text{samples}}$, and to the choice of the distance metric (bit difference in our case), which may affect the clustering results. DBSCAN may also be computationally expensive for a large number of data points. This is less of an issue in our case since it is lower than $256 \times 256$.

We use the Scikit-learn (Pedregosa et al., 2011) implementation of DBSCAN in our experiments.

## C.2 JUST-NOTICEABLE-DIFFERENCE

The maximum change that the human visual system (HVS) cannot perceive is referred to as the Just-Noticeable-Difference (JND) (Krueger, 1989). It is used in image/video watermarking, compression, quality assessment, etc. JND models in the pixel domain directly calculate the JND at each pixel location (i.e., how much pixel difference is perceivable by the HVS).

This section describes in detail the JND map used in Sec. 4.3. It is based on the work of Chou & Li (1995). We use this model for its simplicity, its efficiency, and its good qualitative results. More complex HVS models could also be used if higher imperceptibility is needed (Watson, 1993; Yang et al., 2005; Zhang et al., 2008; Jiang et al., 2022). The JND map takes into account two characteristics of the HVS, namely the luminance adaptation (LA) and the contrast masking (CM) phenomena. We follow the same notations as Wu et al. (2017); Fernandez et al. (2023b), and consider images that are in the range $[0, 255]^{3 \times 256 \times 256}$.

**Luminance masking.** Luminance masking refers to the phenomenon where the HVS is less sensitive to distortions in bright regions of an image. We denote by $K_{lum}$ the luminance kernel, and $x$ the input image. The local background luminance of the image is:

$$B(x)(i,j) = \frac{1}{32} \sum_{k,l \in [-2,2]^2} K_{lum}(k,l) \cdot x(i+k, j+l), \text{ with } K_{lum} = \begin{bmatrix} 1 & 1 & 1 & 1 & 1 \\ 1 & 2 & 2 & 2 & 1 \\ 1 & 2 & 0 & 2 & 1 \\ 1 & 2 & 2 & 2 & 1 \\ 1 & 1 & 1 & 1 & 1 \end{bmatrix}. \quad (6)$$

These luminance values are then post-processed as follows to account for the non-linear response of the HVS:

$$LA(x)(i,j) = \begin{cases} 17 \cdot \left(1 - \sqrt{\frac{B(x)(i,j)}{127} + \epsilon}\right) + 3, & B(x)(i,j) \leq 127 \\ \frac{3}{128} \cdot (B(x)(i,j) - 127) + 3, & B(x)(i,j) > 127, \end{cases} \quad (7)$$

where $\epsilon$ is a small positive value to ensure differentiability during back-propagation.

**Contrast masking.**    Contrast masking refers to the phenomenon where the HVS is less sensitive to distortions in regions of high contrast. The gradient magnitude is:

$$C(x)(i,j) = \sqrt{\left(\sum_{k,l} K_X(k,l) \cdot x(i+k, j+l)\right)^2 + \left(\sum_{k,l} K_Y(k,l) \cdot x(i+k, j+l)\right)^2}, \quad (8)$$

$$\text{with } K_X = \begin{bmatrix} -1 & 0 & 1 \\ -2 & 0 & 2 \\ -1 & 0 & 1 \end{bmatrix}, K_Y = \begin{bmatrix} -1 & -2 & -1 \\ 0 & 0 & 0 \\ 1 & 2 & 1 \end{bmatrix}, \quad (9)$$

where $K_X$ and $K_Y$ are the horizontal and vertical gradient kernels, and $x$ is the input image. To account for the non-linear response of the HVS, the contrast masking values $CM(x)$ at each pixel location:

$$CM(x) = \frac{16 \cdot C(x)^{2.4}}{C(x)^2 + 26^2}. \quad (10)$$

**Heatmap generation.**    The heatmap generation component of the JND model combines the luminance masking and contrast masking values to produce the JND heatmap:

$$H(x) = LA(x) + CM(x) - \gamma \cdot \min(LA(x), CM(x)), \quad (11)$$

where $\gamma$ is a parameter that controls the trade-off between luminance masking and contrast masking.

For color images, we compute the heatmap from the image's luminance. Then we repeat it over the 3 color channels but with a scaling that differs for each channel: $H_{\text{JND}} = [\alpha_R H, \alpha_G H, \alpha_B H]$, where $(\alpha_R, \alpha_G, \alpha_B) = (1, 1, 2)$, because the human eye is more sensitive to red and green than blue color shifts. This produces slightly more distortion in the blue channel.

## D    IMPLEMENTATION DETAILS & PARAMETERS

### D.1    ARCHITECTURES OF THE WATERMARK EMBEDDER/EXTRACTOR

We hereby detail the modeling choices and architectures of Sec. 4.1. Our models, in particular the embedder, are kept voluntarily small to be fast and usable at scale. For instance the embedder and extractor described bellow have respectively 1.1M and 96M ($\approx$ ViT-Base) parameters. Further work could explore how to build larger models with same throughput to improve the results.

In the following we consider an image $x \in \mathbb{R}^{3 \times H \times W}$ and a message $m \in \{0, 1\}^{n_{\text{bits}}}$ The embedder and extractor operate at resolution $h \times w = 256 \times 256$ (see Sec. 4.1).

**Embedder.**    Our goal is to embed a message in an image in a way that is imperceptible to the human eye. This task is similar in many ways to image compression or image-to-image translation. The most used architectures for this arguably come from the works of Rombach et al. (2022); Esser et al. (2021), which strike a very good balance between image quality and efficiency.

The encoder and decoder are described in Tab. 3 (we refer to the VQGAN paper (Esser et al., 2021) for more details). They mainly consist of residual blocks optionally followed by upsampling or downsampling blocks. For the encoder, we use $m = 4$ residual blocks with output channels $d = 32, 32, 32, d' = 64$, downsampling factor of 2 for the 3 first blocks (leading to a division by $f = 8$ of the edge size of the latent map), and $d_z = 4$. The decoder mirrors the encoder, and we choose $d_{\text{msg}} = n_{\text{bits}} = 32$. The Up block interpolates the activation map to the new size, then applies a 2D convolution with kernel size 3, stride 1 and padding 1. In particular, we choose not to use deconvolution layers (ConvTranspose2D) because of the checkerboard patterns they introduce (Odena et al., 2016). The Down block average-pools the activation map with $2 \times 2$ kernels with stride 2.

Our task does not need a bottleneck, since we are not interested in learning compressed representations. Therefore the autoencoder predicts a signal $\delta \in [-1, 1]^{3 \times H \times W}$ – and not directly the final image – which is added to the image with a residual layer. Another difference is that the message is embedded in the latent space of the encoder, which means that the first layer of the decoder is bigger. As a reminder, the binary message lookup table is $\mathcal{T}_\theta \in \mathbb{R}^{n_{\text{bits}} \times 2 \times d_{\text{msg}}}$. Each bit of the message $m$ is mapped to the embedding $\mathcal{T}_\theta(k, m_k, \cdot) \in \mathbb{R}^{d_{\text{msg}}}$, depending on its position $k \in \{1, \ldots, n_{\text{bits}}\}$ and its value $m_k \in \{0, 1\}$, then repeated, which yields $z_{\text{msg}} = \text{repeat}\left(1/n_{\text{bits}} \sum_{k=1}^{n_{\text{bits}}} \mathcal{T}_\theta(k, m_k, \cdot)\right) \in \mathbb{R}^{d_{\text{msg}} \times 32 \times 32}$.

Table 3: High-level architecture of the encoder and decoder of the watermark embedder $\text{emb}_\theta$. The design of the networks follows the architecture presented by Ho et al. (2020); Esser et al. (2021); Rombach et al. (2022).

| Encoder | Decoder |
|---|---|
| $x \in \mathbb{R}^{3 \times H \times W}, m \in \{0,1\}^{n_{\text{bits}}}$ | $(z, z_{\text{msg}}) \in \mathbb{R}^{(d_z + d_{\text{msg}}) \times 32 \times 32}$ |
| Interpolation, Conv2D $\to \mathbb{R}^{d \times 256 \times 256}$ | Conv2D $\to \mathbb{R}^{d' \times 32 \times 32}$ |
| $m \times \{$ Residual Block, Down Block $\} \to \mathbb{R}^{d' \times 32 \times 32}$ | Residual Block $\to \mathbb{R}^{d' \times 32 \times 32}$ |
| Residual Block $\to \mathbb{R}^{d' \times 32 \times 32}$ | Non-Local Block $\to \mathbb{R}^{d' \times 32 \times 32}$ |
| Non-Local Block $\to \mathbb{R}^{d' \times 32 \times 32}$ | Residual Block $\to \mathbb{R}^{d' \times 32 \times 32}$ |
| Residual Block $\to \mathbb{R}^{d' \times 32 \times 32}$ | $m \times \{$ Residual Block, Up Block $\} \to \mathbb{R}^{d \times 256 \times 256}$ |
| GroupNorm, Swish, Conv2D $\to \mathbb{R}^{d_z \times 32 \times 32}$ | GroupNorm, Swish, Conv2D $\to \mathbb{R}^{3 \times 256 \times 256}$ |
| $\mathcal{T}_\theta(m)$, Repeat $\to \mathbb{R}^{d_{\text{msg}} \times 32 \times 32}$ | TanH, Interpolation $\to [-1, 1]^{3 \times H \times W}$ |

Table 4: High-level architecture of the encoder and decoder of the watermark extractor $\text{ext}_\theta$. The design of the networks follows the architecture presented by Zheng et al. (2021); Kirillov et al. (2023).

| Image encoder (ViT) | Pixel decoder (CNN) |
|---|---|
| $x \in \mathbb{R}^{3 \times H \times W}$ | $z \in \mathbb{R}^{d' \times 16 \times 16}$ |
| Interpolation $\to \mathbb{R}^{3 \times 256 \times 256}$ | $m' \times \{$ Residual Block, Up Block $\} \to \mathbb{R}^{d'' \times 256 \times 256}$ |
| Patch Embed (Conv2D), Pos. Embed $\to \mathbb{R}^{d \times 16 \times 16}$ | Linear $\to \mathbb{R}^{(1+n_{\text{bits}}) \times 256 \times 256}$ |
| $m \times \{$ Transformer Block $\} \to \mathbb{R}^{d \times 16 \times 16}$ | Sigmoid (optional) $\to \mathbb{R}^{(1+n_{\text{bits}}) \times 256 \times 256}$ |
| LayerNorm, GELU, Conv2D $\to \mathbb{R}^{d' \times 16 \times 16}$ | Interpolation $\to \mathbb{R}^{(1+n_{\text{bits}}) \times H \times W}$ |

**Extractor.** Our goal is to detect and extract the message from an image, at the pixel level. This task is similar to image segmentation. Our extractor is based on a vision transformer (ViT) (Dosovitskiy, 2020) followed by a pixel decoder, as commonly done in the literature (Kirillov et al., 2023; Zheng et al., 2021; Oquab et al., 2023).

The architecture of the extractor is detailed in Tab. 4. The encoder is a ViT which consists of a series of attention blocks to process the image's patches into a high-dimensional feature space. We use the ViT-Base architecture (86M parameters), with patch size 16 (Dosovitskiy, 2020), with $d = d' = 768$. The patch embeddings are then upscaled by the pixel decoder to predict the segmentation masks and messages for every pixel. The latter uses $m' = 3$ Up blocks which upsample (bilinearly) the activation map by factors $4, 2, 2$ respectively (total upscale factor is 16, which is the patch size used in the ViT encoder). Each Up block is followed by a Conv2D with kernel size of 3 and stride of 1, a LayerNorm, and a GELU activation. The number of channels output by the convolution is its number of input channels divided by the upsampling factor of the Up block that precedes. We obtain a latent map of shape $(d'' = d/16, 256, 256)$, which is mapped to $(1 + n_{\text{bits}})$-dimensional pixel features by a linear layer. Finally, a Sigmoid layer scales the outputs to $[0, 1]$ (this is in fact only done at inference, since the training objective implicitly applies it in PyTorch).

## D.2 MASK GENERATION

We follow the protocol of LaMa (Suvorov et al., 2022), and generate most of our masks used during the two phases of training by adapting the authors' code github.com/advimman/lama. Figure 6 shows some qualitative examples. More specifically, a diverse set of random masks are used:

- Box-shaped masks are defined with a margin of 10 pixels and bounding box sizes width and height randomly chosen, ranging from 30 to 100 pixels. They can be generated multiple times (1 to 3 times) per image.

- Full-image masks cover the entire image.

- Segmentation-based masks are segments corresponding to object boundaries or specific areas within an image. We use the masks of the COCO dataset, and either select the union of all masks with probability 0.5, or the union of a random number of objects.

- Irregular masks are random brush strokes, characterized by parameters such as maximum and minimum angles (up to 4 degrees), lengths and widths (ranging from 20 to 50 pixels). The brush strokes can be applied between 1 to 5 times per image.

During the first training phase (described in Sec. 4.2), we use irregular, box-shaped, full-image, and segmentation-based masks, each with a probability of 0.25. Additionally, there is a 50% chance of inverting any of the masks.

During the second training phase (described in Sec. 4.3), we use the same masks except that:

- for "box-shaped masks", with probability 0.5, instead of outputting a single mask as in the first phase, we output a random number from 1 to 3 of non overlapping rectangles;

- for "segmentation-based masks", with probability 0.5, instead of outputting a single mask as in the first phase, we output a random number from 1 to 3 of segmented objects.

When randomly selected during this post-training phase, both types of multiple masks cannot be inverted and they are used to hide different watermarks within the same image.

## D.3   WATERMARKING METHODS

We follow the evaluation setup used by Fernandez et al. (2023a). For HiDDeN (Zhu et al., 2018), we use the model available in github.com/facebookresearch/stable_signature and modulate the output with the same JND mask. For DCTDWT, we use the implementation of github.com/ShieldMnt/invisible-watermark (the one used in Stable Diffusion). For SSL Watermark (Fernandez et al., 2022) and FNNS (Kishore et al., 2022) the watermark is embedded by optimizing the image, such that the output of a pre-trained model is close to the given key. SSL Watermark uses a model pre-trained with DINO (Caron et al., 2021), while FNNS uses the HiDDeN extractor used in all our experiments, and not SteganoGan (Zhang et al., 2019a) as in the original paper. We optimize the distortion image for 10 iterations, and modulate it with the same JND mask. This avoids visible artifacts and gives a PSNR comparable to our method ($\approx 38$dB). For TrustMark (Bui et al., 2023a), we use the "C variant" with 40 bits in the official implementation github.com/adobe/trustmark, as it achieves a similar PSNR.

## D.4   TRANSFORMATIONS

Transformations seen at training time and evaluated in Sec. 5 simulate common image processing steps. We categorize them into different groups: *valuemetric*, which change the pixel values; *geometric*, which modify the image's geometry; *splicing*, which add watermarked areas inside

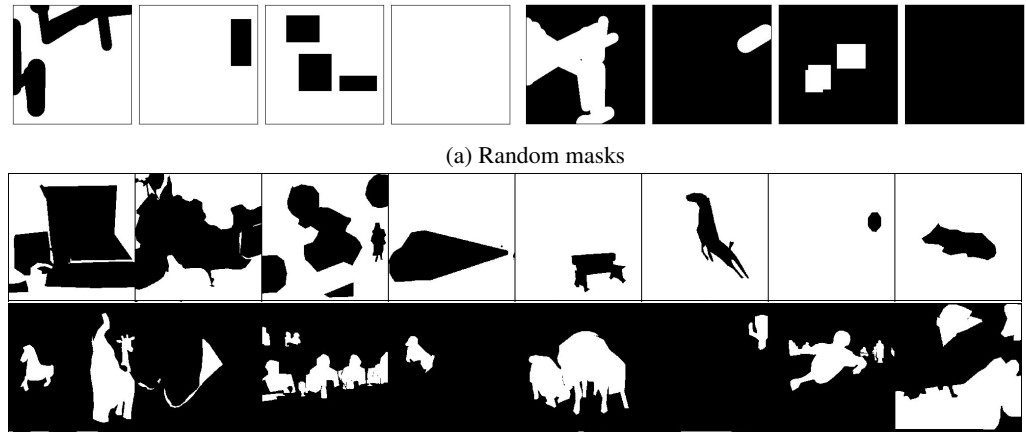

(a) Random masks

(b) Segmentation masks

Figure 6: Examples of masks used during training. Only the white areas of the image end up being watermarked. (a) Random masks (irregular, rectangles, inverted, full, null or inverted). (b) Segmentation masks created from the union of COCO's segmentation masks.

Table 5: Illustration of transformations evaluated in Sec. 5.

the image; and *inpainting*, which fill masked areas with plausible content. The geometric and valuemetric transformations are displayed in Tab. 5 and detailed in the following table:

| Transformation | Type | Parameter | Training | Evaluation |
|---|---|---|---|---|
| Brightness | Valuemetric | from torchvision | Random between 0.5 and 2.0 | 1.5 and 2.0 |
| Contrast | Valuemetric | from torchvision | Random between 0.5 and 2.0 | 1.5 and 2.0 |
| Hue | Valuemetric | from torchvision | Random between -0.1 and 0.1 | -0.1 and 0.1 |
| Saturation | Valuemetric | from torchvision | Random between 0.5 and 2.0 | 1.5 and 2.0 |
| Gaussian blur | Valuemetric | kernel size $k$ | Random odd between 3 and 17 | 3 and 17 |
| Median filter | Valuemetric | kernel size $k$ | Random odd between 3 and 7 | 3 and 7 |
| JPEG | Valuemetric | quality $Q$ | Random between 40 and 80 | 50 and 80 |
| Horizontal flip | Geometric | NA | | |
| Crop | Geometric | edge size ratio $r$ | Random between 0.33 and 1.0 | 0.33 and 0.5 |
| Resize | Geometric | edge size ratio $r$ | Random between 0.5 and 1.5 | 0.5 |
| Rotation | Geometric | angle $\alpha$ | Random between -10 and 10 | -10 and 10 |
| Perspective | Geometric | distortion scale $d$ | Random between 0.1 and 0.5 | 0.1 and 0.5 |

For crop and resize, each new edge size is selected independently, which means that the aspect ratio can change (because the extractor resizes the image). Moreover, an edge size ratio of $0.33$ means that the new area of the image is $0.33^2 \approx 10\%$ times the original area. For brightness, contrast, saturation, and sharpness, the parameter is the default factor used in the PIL and Torchvision (Marcel & Rodriguez, 2010) libraries. For *splicing*, we crop a random area of the image and paste it back at the same location on the original image or on a different background image.

For evaluation (Sec. 5.3) we select some transformations from the list above and apply them with the parameters given in the table. We chose these parameters high enough to have pronounced effects on the robustness of the watermark. In practice, they are quite strong and would not not be encountered often in real-world scenarios. Additionally we evaluate the robustness against *inpainting* which is not seen at training time. We use LaMa (Suvorov et al., 2022) to modify masked areas in the image conditioned on the rest of the image.

## D.5 LOCALIZATION EXPERIMENTS

Figure 7 gives an example of how the evaluation for watermark localization is performed in Sec. 5.4, for the "crop 0.25" augmentation. The mask used to place the watermark (second image) is the same for each image with an area covering from 10% to 100% of the image. We perform a 25% crop of the upper left corner. After that, we resize the image to its original size. Note that after this augmentation, the watermarked area still recovers the same proportion of the crop (by design, as the watermarked area was originally centered).

### D.6 MULTIPLE WATERMARKS

Figure 8 shows an example of how the evaluation for multiple watermarks is performed in Sec. 5.5. After watermarking 5 areas of the image that are 10% each with different messages, the extractor detects several watermarked areas and outputs one message for each.

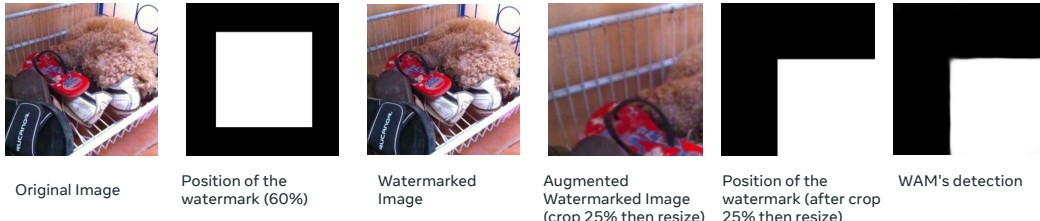

Figure 7: Experimental protocol for the evaluation of watermark localization as performed in Sec. 5.4.

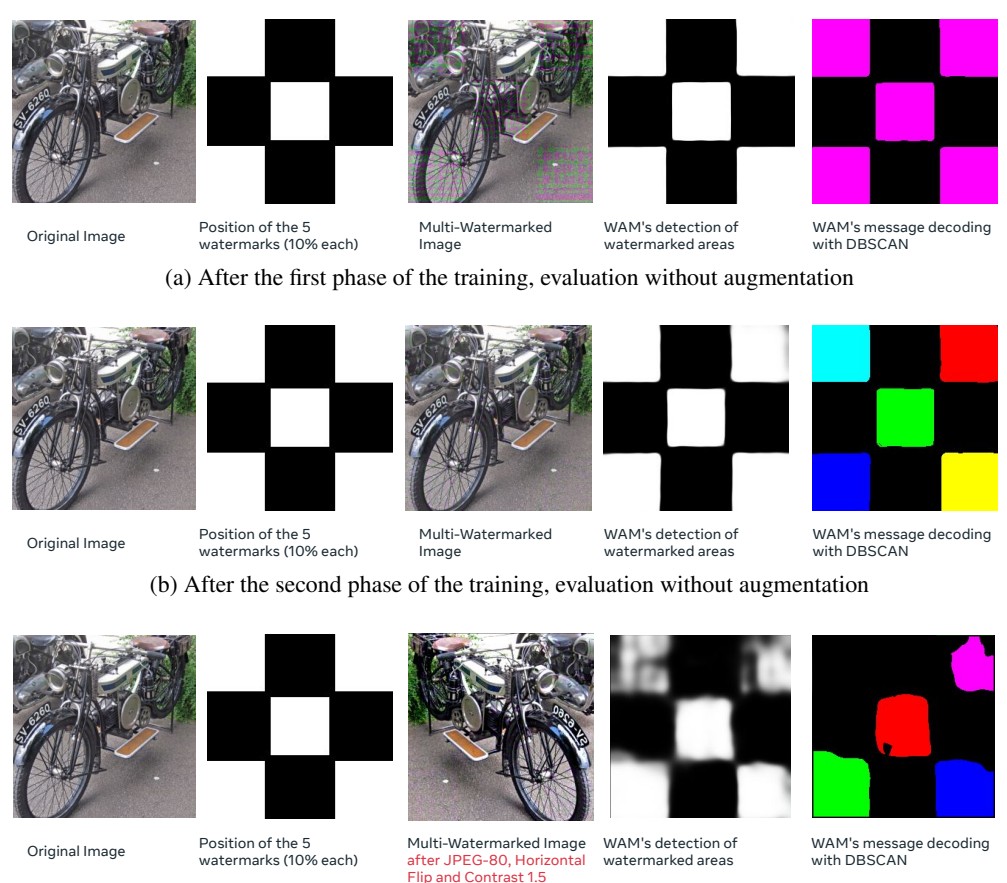

(a) After the first phase of the training, evaluation without augmentation

(b) After the second phase of the training, evaluation without augmentation

(c) **Failure:** after the second phase of the training, with JPEG-80, horizontal flip and contrast 1.5

Figure 8: Evaluation protocol for multiple watermark extraction as described in Sec. 5.5. The masks used to place the different watermarks (second image) are the same for each image. We then evaluate the bit accuracy across all discovered messages and their ground truth. Different colors represent different detected clusters for each image, but the color scheme is not coherent between images.

Table 6: Detection results for WAM and generation-time watermarking methods, on 1k negative (non-watermarked), and 1k positive (watermarked), possibly edited, images. AUC refers to the Area Under the ROC curve, TPR@$10^{-2}$ is the TPR at FPR= $10^{-2}$. Stable Signature (Fernandez et al., 2023a) embeds a 48-bit message and uses the bit accuracy as score, while Tree-Ring (Wen et al., 2023) and WAM output a detection score. WAM is competitive with watermarking methods for generative models (although the latter offer noteworthy advantages).

| | AUC | | | TPR@$10^{-2}$ | | | TPR@$10^{-4}$ | | |
|---|---|---|---|---|---|---|---|---|---|
| | Stable Sig. | Tree-Ring | WAM | Stable Sig. | Tree-Ring | WAM | Stable Sig. | Tree-Ring | WAM |
| None | 1.00 | 1.00 | 1.00 | 1.00 | 1.00 | 1.00 | 1.00 | 1.00 | 1.00 |
| Valuemetric | 0.94 | 1.00 | 1.00 | 0.90 | 1.00 | 1.00 | 0.90 | 1.00 | 1.00 |
| Geometric | 1.00 | 0.91 | 1.00 | 0.97 | 0.56 | 1.00 | 0.87 | 0.43 | 1.00 |
| Comb. | 1.00 | 0.75 | 1.00 | 0.99 | 0.05 | 1.00 | 0.99 | 0.01 | 1.00 |
| Splicing | 0.97 | 0.72 | 1.00 | 0.90 | 0.00 | 1.00 | 0.81 | 0.00 | 0.00 |

# E  ADDITIONAL RESULTS

## E.1  COMPARISON WITH WATERMARKING METHODS FOR GENERATIVE MODELS

So far, we only considered general watermarking methods that apply the watermark on existing images. We now compare WAM to methods specific for LDM, which watermark at generation time. We generate 1k images from text prompts with Stable Signature (Fernandez et al., 2023a), Tree-Ring (Wen et al., 2023) and without watermarking. We also generate a set of watermarked images with WAM from this last set. For each method, we retrieve a detection score for both the watermarked set of images and the non-watermarked images, that we use to compute the Receiver Operating Characteristic (ROC) curve and the Area Under the Curve (AUC) for each method. In the case of Stable Signature, we embed a 48-bit binary message and the score is the bit accuracy with this message, while Tree-Ring and WAM directly output a detection score.

The results are shown in Tab. 6. We observe that WAM overall obtains better performance. However, the watermarking methods for generative models are still competitive and offer important advantages, namely, the possibility to open-source the model for Stable Signature, and the possibility to watermark with virtually no image degradation as well as very strong robustness against valuemetric transformations for Tree-Ring. Note that the results are not perfectly apple-to-apple since the efficiency of the extractor is different between methods: Tree-Ring requires to inverse the diffusion noise which is considerably heavier, while Stable Signature uses a very small extractor ($\approx$ 100k parameters).

## E.2  HYPER PARAMETER SEARCH FOR DBSCAN

In Section 5.5, the DBSCAN algorithm is used to decode multiple messages from each image, with parameters set to $\varepsilon = 1$ and $\text{min}_{\text{samples}} = 1000$. Under the same experimental setting, Figure 9 displays the results of a hyperparameter search conducted under two scenarios: images that underwent horizontal flipping and a contrast adjustment with a parameter of 1.5 on the left, and the same

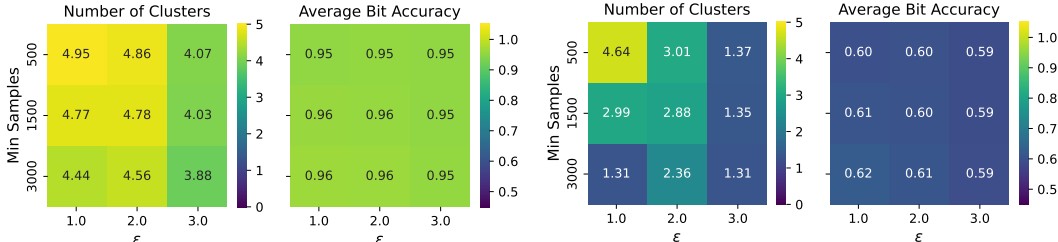

(a) After horizontal hlip and contrast adjustment of 1.5. The overall mIoU is at 85%.   (b) After horizontal hlip, contrast adjustment of 1.5and JPEG-80. The overall mIoU is at 59%.

Figure 9: DBSCAN hyperparameter search in the same setting as in Sec. 5.5, under two different types of augmentations. Qualitative examples are shown in Fig. 8.

Table 7: Full decoding results on the COCO validation set, used for the aggregated results presented in Tab. 2a of Sec 5.3. "Combination" corresponds to JPEG-80, Brightness 1.5 and Crop 50% all on the same image, and was not part of the evaluation of Sec. 5.3. TPR is for a threshold on the detection score $s_{\text{det}}$ such that FPR $= 0.04\%$. The bit accuracy is computed as described in Eq. 2.

| | HiDDeN | | DCTDWT | | SSL | | FNNS | | TrustMark | | WAM | |
|---|---|---|---|---|---|---|---|---|---|---|---|---|
| | TPR / Bit acc. | | TPR / Bit acc. | | TPR / Bit acc. | | TPR / Bit acc. | | TPR / Bit acc. | | TPR / Bit acc. | |
| None | 76.0 | 95.5 | 77.1 | 91.4 | 99.9 | 100.0 | 99.6 | 99.9 | 99.8 | 99.9 | 100.0 | 100.0 |
| Crop (20%) | 66.5 | 93.3 | 0.1 | 50.2 | 3.3 | 68.8 | 98.5 | 99.6 | 2.1 | 50.3 | 99.7 | 88.8 |
| Crop (50%) | 71.0 | 94.6 | 0.0 | 50.4 | 7.0 | 73.4 | 99.2 | 99.8 | 1.9 | 60.9 | 99.6 | 95.9 |
| Rot (10%) | 7.1 | 80.8 | 0.0 | 50.7 | 11.2 | 78.1 | 76.0 | 94.9 | 3.0 | 56.4 | 97.7 | 77.0 |
| Perspective (0.1) | 65.7 | 93.6 | 0.0 | 51.8 | 15.6 | 80.0 | 98.0 | 99.6 | 77.8 | 96.7 | 100.0 | 99.8 |
| Perspective (0.5) | 32.3 | 89.1 | 0.1 | 50.0 | 0.8 | 61.3 | 51.6 | 91.3 | 2.6 | 50.8 | 100.0 | 96.0 |
| Horizontal Flip | 0.1 | 54.3 | 0.0 | 50.2 | 32.7 | 86.2 | 0.1 | 56.2 | 99.8 | 99.9 | 100.0 | 100.0 |
| Brightness (1.5) | 85.6 | 96.9 | 16.1 | 60.9 | 90.9 | 97.7 | 99.1 | 99.7 | 83.2 | 97.1 | 100.0 | 100.0 |
| Brightness (2.0) | 83.2 | 96.2 | 0.1 | 49.4 | 67.8 | 91.9 | 97.2 | 99.0 | 58.8 | 92.2 | 100.0 | 99.9 |
| Contrast (1.5) | 74.1 | 95.4 | 20.4 | 63.4 | 92.9 | 98.2 | 98.9 | 99.7 | 77.1 | 96.2 | 100.0 | 100.0 |
| Contrast (2.0) | 67.3 | 94.3 | 0.6 | 49.4 | 66.0 | 92.4 | 97.0 | 99.4 | 52.1 | 90.8 | 100.0 | 99.9 |
| Hue (-0.1) | 9.0 | 77.4 | 29.5 | 66.7 | 98.0 | 99.4 | 96.7 | 99.2 | 99.3 | 99.9 | 100.0 | 100.0 |
| Hue (+0.1) | 19.3 | 84.0 | 59.4 | 81.2 | 97.7 | 99.3 | 98.5 | 99.5 | 99.5 | 99.9 | 100.0 | 100.0 |
| Saturation (1.5) | 80.6 | 96.2 | 21.3 | 61.7 | 99.7 | 99.9 | 99.5 | 99.9 | 99.0 | 99.8 | 100.0 | 100.0 |
| Saturation (2.0) | 81.7 | 96.5 | 0.3 | 47.4 | 98.3 | 99.5 | 99.4 | 99.8 | 96.8 | 99.5 | 100.0 | 100.0 |
| Median filter (3) | 74.6 | 94.9 | 0.9 | 53.2 | 82.8 | 96.5 | 99.4 | 99.9 | 99.7 | 99.9 | 100.0 | 100.0 |
| Median filter (7) | 23.4 | 83.0 | 0.4 | 53.0 | 20.4 | 80.7 | 61.1 | 90.2 | 99.4 | 99.9 | 100.0 | 100.0 |
| Gaussian Blur (3) | 47.1 | 88.0 | 41.7 | 77.0 | 97.2 | 99.1 | 87.3 | 95.5 | 99.8 | 99.9 | 100.0 | 100.0 |
| Gaussian Blur (17) | 0.1 | 51.2 | 0.0 | 49.8 | 3.8 | 69.6 | 0.0 | 50.5 | 98.7 | 99.8 | 100.0 | 99.8 |
| JPEG (50) | 11.2 | 77.3 | 0.0 | 49.8 | 5.1 | 72.5 | 34.5 | 86.9 | 99.1 | 99.8 | 99.9 | 99.0 |
| JPEG (80) | 32.8 | 86.8 | 0.1 | 50.5 | 66.1 | 92.6 | 80.5 | 95.4 | 99.6 | 99.9 | 100.0 | 99.9 |
| Proportion (10%) | 0.8 | 65.9 | 1.1 | 56.9 | 0.6 | 61.8 | 36.3 | 88.2 | 2.0 | 58.6 | 99.9 | 94.2 |
| Collage (10%) | 0.9 | 70.1 | 0.5 | 62.8 | 0.1 | 55.9 | 41.1 | 88.7 | 1.3 | 55.7 | 100.0 | 96.5 |
| Combination | 44.7 | 88.7 | 0.0 | 49.9 | 1.2 | 62.8 | 87.7 | 96.1 | 1.7 | 58.8 | 99.2 | 87.2 |

transformations followed by JPEG compression at a quality level of 80 on the right. In the absence of JPEG compression, WAM effectively recovers the correct number of messages with satisfactory bit accuracies. However, when JPEG compression is applied on top, WAM encounters difficulties. For both scenarios, smaller values of $\varepsilon$ and $\min_{\text{samples}}$ yielded the most favorable outcomes. These findings guided the selection of the aforementioned values for the primary evaluation in sec. 5.5.

### E.3    DETAILED ROBUSTNESS RESULTS

Table 7 shows the detailed results for all the transformations. We observe that WAM handles many of them, although the performance decreases as they get stronger (e.g., for the combination). Figure 10 shows several examples of segmentation masks predicted after LaMa inpainting applied to both COCO and DIV2K datasets, consistent with the evaluation settings described in Sec. 5.3.

## F    QUALITATIVE EXAMPLES

We show examples of images from the COCO dataset in Fig. 11 and for DIV2k in Fig. 12. We observe that the watermark is imperceptible to the human eye, primarily due to the JND map used to mask the watermark in areas where the human visual system is less sensitive. With enough attention, it is however possible to see it in some images, especially in very white or very dark areas (e.g., the boat in the last row of Fig. 11) and in the fur of the animals (e.g., the wolf and the penguin of Fig. 12). We hypothesize that it comes from the fact that the JND map only accounts for the cover image where the watermark signal $\delta$ is added, and not for $\delta$ itself. However, repetitive and regular patterns are perceptible, and, when the bright or textured areas are big enough, this becomes noticeable. We believe that further work could improve the imperceptibility of the watermark, for instance by using more complex HVS models (Watson, 1993) or by using a regularization on the watermark signal to remove repetitive or structured patterns.

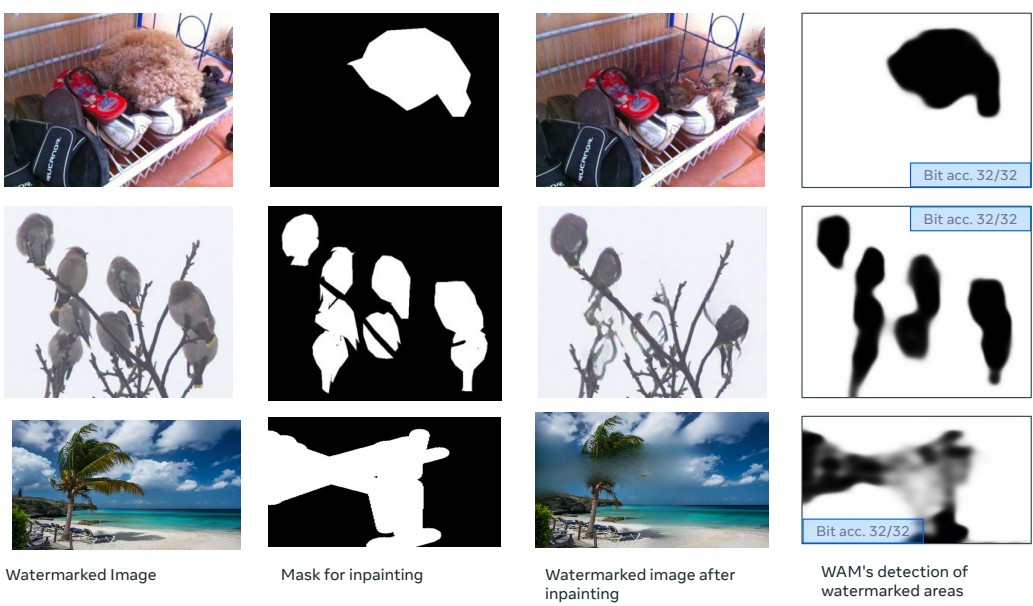

Figure 10: Examples of WAM's detection after inpainting with LaMa (Suvorov et al., 2022), with the experimental set-up detailed in Sec. 5.3. The first two lines are with images from the validation set of COCO (using the union of the segmentation masks), while the third line is with an image from the DIV2k dataset, with a random mask.

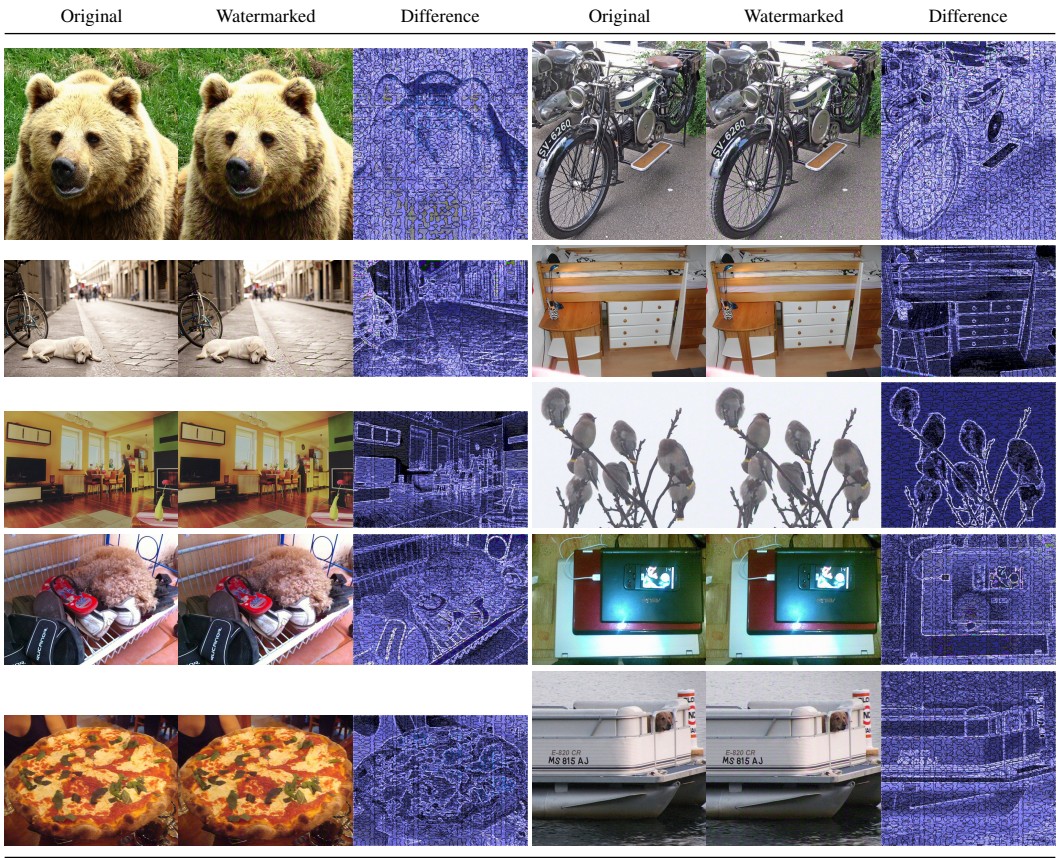

Figure 11: Qualitative results on the validation set of MS-COCO, at various resolutions and for a 32-bits message. The difference image is displayed as $10 \times \text{abs}(x_\text{m} - x)$.

| Original | Watermarked | Difference |
| --- | --- | --- |

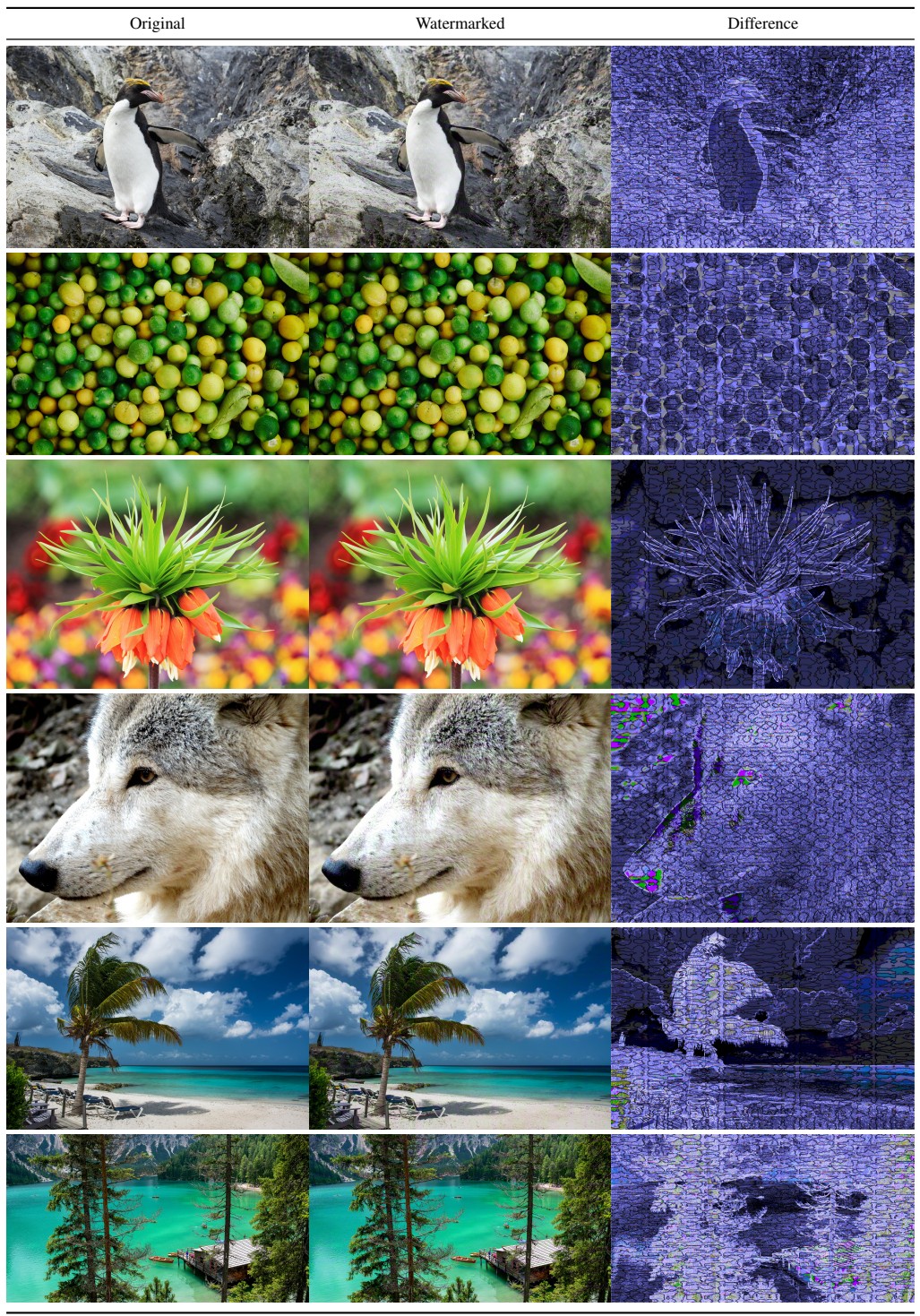

Figure 12: Qualitative results on images from DIV2k, at higher resolution and for a 32-bits message. The difference image is displayed as $10 \times \text{abs}(x_m - x)$.

Table 8: Bit accuracy for MBRS, CIN, and WAM under different augmentations.

| Method | None | Geometric | Valuemetric | Splicing | PSNR/SSIM/LPIPS |
|--------|------|-----------|-------------|----------|-----------------|
| MBRS | 100.0 | 50.4 | 99.8 | 65.31 | 38.8/0.99/0.07 |
| CIN | 100.0 | 50.3 | 100.0 | 96.47 | 38.3/0.99/0.08 |
| WAM (ours) | 100.0 | 91.8 | 100.0 | 95.30 | 38.3/0.99/0.04 |

## G  ADDITIONAL DETAILS AND EXPERIMENTS AFTER REBUTTAL

This section provides further insights and experimental results following discussions with the reviewers. We address the operating point of WAM and additional experiments on robustness evaluations.

### G.1  OPERATING POINT: ARE 32 BITS ENOUGH?

Our primary objective is to achieve extreme robustness against typical internet/social network image sharing, which necessitates a trade-off with capacity. WAM demonstrates significant robustness against geometric augmentations compared to other methods (see Table 2a). It can localize watermarks and extract a 32-bit message from just 10% of a 256×256 image, which is the operating point we have chosen. Alternative configurations could be considered.

WAM offers three additional functionalities: localization, the ability to hide multiple 32-bit messages, and detection separate from decoding. The second functionality indicates a higher true capacity than the 32 bits, and the third ensures that 32 bits are sufficient for many applications. To obtain larger payloads, one could theoretically extend the training. Increasing the localization constraint from 10% to 20% of 256×256 or training on 512×512 resolution while maintaining a 10% target area, or lightening the robustness constraints during training could also help.

### G.2  CHOICE OF BASELINES FOR COMPARISON

We compared methods present in the paper as they constitute a solid baseline. Our goal was not to demonstrate that WAM is more robust than all methods in every setting, but rather to show that it performs well for classical watermarking while introducing new capabilities. We argue that MBRS (Jia et al., 2021), FIN (Fang et al., 2023), and CIN (Ma et al., 2022) are not robust to crops, indicating a different operating point. Our focus is on decoding the image even from small watermarked areas.

As asked by two reviewers, we add comparisons to CIN and MBRS. Specifically, MBRS has a 256-bit payload. Like other methods, we encode 16 bits for detection and 32 bits for decoding. We encode this 16+32 = 48-bit message 5 times, occupying 240 bits. At decoding time, we average the outputs of the decoder every 48 bits (soft majority vote) and output the decoded 48-bit message. CIN, on the other hand, only handles 30 bits. We show the bit accuracy for the first 30 bits of the message for CIN and do not perform detection (a favorable setup for CIN). For both methods, we scale the distortion to achieve a similar PSNR for all models. Results are shown in Table 8. This is the exact same evaluation set-up as for Table 2a and Table 1 on COCO.

### G.3  ROBUSTNESS AGAINST NEURAL PURIFICATION

**Diffusion based purification.**  We evaluate robustness against diffusion-based purification in Table 9, acknowledging its challenge for all watermarks. While AI-based purification is a significant challenge for image watermarks, it is not the focus of our work, and we share the same limitations as others. It is efficient to remove post-hoc watermarks, but it requires a good adversary, and is less likely to happen "in the wild". However, Table 2a shows WAM's robustness against strong inpainting, which is more likely to happen by a typical non adversarial user.

**Neural encoders.**  We conducted new experiments to assess WAM's performance against VAE attacks, as done by Fernandez et al. (2023a). We measured bit accuracy for a 32-bit message, consistent with Table 2a. WAM remains robust until the VAE significantly compresses the image

Table 9: Robustness of various methods against DiffPure purification and VAE attack at different PSNR levels between original and recovered image.

| Method | DiffPure (PSNR) | | | | VAE Attack (PSNR) | | | |
|---|---|---|---|---|---|---|---|---|
| | 30.1 | 28.4 | 27.0 | 24.9 | 32.9 | 32.4 | 28.6 | 25.5 |
| HiDDeN | 88.6 | 65.2 | 56.2 | 51.6 | 82.8 | 80.6 | 56.9 | 50.7 |
| DWTDCT | 80.4 | 48.6 | 49.0 | 49.5 | 72.4 | 63.1 | 49.7 | 49.5 |
| SSL | 99.2 | 63.0 | 54.2 | 52.0 | 98.4 | 97.6 | 66.8 | 51.6 |
| FNNS | 97.9 | 64.1 | 53.0 | 50.1 | 97.9 | 95.1 | 55.9 | 50.2 |
| TrustMark | 100.0 | 98.3 | 74.9 | 54.3 | 100.0 | 100.0 | 99.9 | 98.1 |
| MBRS | 100.0 | 99.5 | 85.1 | 64.8 | 100.0 | 100.0 | 100.0 | 97.7 |
| CIN | 100.0 | 100.0 | 82.3 | 61.9 | 98.1 | 100.0 | 99.7 | 67.6 |
| WAM (ours) | 100.0 | 99.4 | 71.3 | 49.1 | 100.0 | 99.8 | 98.6 | 53.7 |

(PSNR$\approx$25 dB). In this scenario, only MBRS and TrustMark maintain robustness, but they lack resilience against geometric transformations, representing a different trade-off choice.

## G.4 ROBUSTNESS/IMPERCEPTIBILITY TRADE-OFF

The robustness/imperceptibility trade-off can be controlled at encoding time by varying the distortion factor $\alpha_{JND}$ which scales the watermark distortion added to the image. Even if trained for a specific value (2 in our case), a different value can be used at inference to enhance robustness (albeit with increased visibility). Table 10 illustrates the bit-accuracy of a 32-bit message for different values of $\alpha_{JND}$ (from 1 to 3) against a combination of JPEG-80, center 50% crop + resize, and brightness 1.5. This approach allows for a flexible trade-off between robustness and imperceptibility, providing users with the ability to adjust settings based on specific application needs.

| Method | PSNR | Bit Acc. |
|---|---|---|
| WAM$_{1.0}$ | 44.1 | 71.0 |
| WAM$_{1.5}$ | 40.7 | 84.3 |
| WAM$_{2.0}$ | 38.3 | 88.8 |
| WAM$_{2.5}$ | 36.4 | 90.8 |
| WAM$_{3.0}$ | 34.8 | 91.9 |

Table 10: Bit accuracy of a 32-bit message for different values of $\alpha_{JND}$ after strong post processing, demonstrating the robustness/imperceptibility trade-off.

## G.5 DETECTION OF OVERLAPPING WATERMARKS

We demonstrate that WAM can still detect the presence of a watermark even if two watermarks overlap entirely on the whole image with different messages. This highlights the advantage of having decoding separated from detection. Table 11 shows the detection accuracy for one and two overlapping watermarks on one hundred COCO validation images.

| Condition | Detection accuracy |
|---|---|
| One watermark | 100/100 |
| Two overlapping watermarks | 100/100 |

Table 11: Detection accuracy of WAM for one and two overlapping watermarks.

In summary, the additional experiments and analyses presented in this appendix further validate the robustness and versatility of WAM in various challenging scenarios. We appreciate the reviewers' feedback, which has helped us enhance the clarity and depth of our work.

