# OpenReview forum: "Watermark Anything With Localized Messages"
_ICLR.cc/2025/Conference — ICLR 2025 Poster_

### Official Review · Reviewer_UTj6 · 2024-10-20

**Soundness:** 3
**Presentation:** 2
**Contribution:** 2
**Rating:** 3
**Confidence:** 4

**Summary:**

This work proposes a deep-learning-based image watermarking method, dubbed WAM. The encoder of WAM can watermark images with multiple watermarks/32-bit messages and the decoder of WAM can output a detection/segmentation map that identifies the regions containing watermarks and those that do not (called detection) and also a message map that presents the message in each watermarked area (called decoding).

**Strengths:**

1. The proposed method is able to embed multiple watermarks/32-bit messages
2. The method can output a detection/segmentation map that identifies the regions containing watermarks and those that do not

**Weaknesses:**

1. I am unclear on how the model achieves multiple watermark encodings. If I understand correctly, according to Figure 2, the model does not accept a mask input during the inference stage. In that case, how does it inject different messages into different regions? This makes Figure 1 somewhat misleading, as the first two steps suggest that the message can be precisely controlled to be encoded into specific objects. Could the authors provide further clarification on this point?

2. Could you clarify how the mask for the watermark's position is inputted into WAM in Figure 7 of the Appendix? Additionally, could you explain why this specific position was chosen for the localization experiment?

3. In lines 226-232, if I am not mistaken, this method appears to be identical to the method proposed by Bui et al., rather than merely 'a similar approach'. Please provide proper credit.

4. Regarding the claim in lines 304-305, it is unclear why applying JND would make training easier compared to using perceptual loss. Could the authors provide evidence or reasoning to support this claim?   'Applying JND only in the second training phase makes training easier than previous methods relying on “perceptual” or “contradictory” losses.'

5. I believe the reason WAM outperforms the baselines in terms of bit accuracy for splicing is simply because WAM incorporates splicing during training, whereas the other models do not. If the other models were to apply splicing as an augmentation or attack during training, would WAM still outperform them in terms of bit accuracy?

6. Typically, the TPR is reported at a fixed FPR, as seen in Tree Ring. Is the approach in this paper, reporting TPR at varying FPRs, fair and appropriate?

7. I am not entirely convinced of the necessity of embedding multiple messages separately into a single image. If the embedding capacity is large enough (e.g., 100 bits), users could already encode several different messages within a single bit stream. Could the authors provide some practical scenarios where embedding multiple separate messages would be particularly beneficial?

8. What happens if the position masks of multiple watermarks overlap? Additionally, for the watermark to be successfully extracted, what is the minimum proportion the watermark mask should cover relative to the overall image size? How many watermarks can be embedded in a single image at most?

9. A small typo on line 407, 'hide messages or arbitrary length' should be 'hide messages of arbitrary length'?

10. Will the source code and pretrained model be released for the community?

11. Is WAM robust to the new type of regeneration attacks [1], like diffusion-based regeneration?

---
[1] Invisible Image Watermarks Are Provably Removable Using Generative AI

---

I hope I have understood the method correctly. If there are any misunderstandings leading to wrong judgment, I apologize to the authors. I would be open to adjusting my score if the authors address my concerns during the rebuttal.

**Questions:**

Please refer to the weaknesses

---

> ### Author Response · Authors · 2024-11-19
>
> We thank the reviewer for their valuable feedback and time spent on our paper. We try to answer everything and remain available for further discussion.
>
> > 1. I am unclear on how the model achieves multiple watermark encodings. If I understand correctly, according to Figure 2, the model does not accept a mask input during the inference stage. In that case, how does it inject different messages into different regions? This makes Figure 1 somewhat misleading, as the first two steps suggest that the message can be precisely controlled to be encoded into specific objects. Could the authors provide further clarification on this point?
>
> - This process is detailed in Section 5.5: "This is done by feeding the image several times to WAM’s embedder, then pasting the different watermarked areas onto the original image." So yes indeed  the model does not accept a mask input during the inference stage. We will clarify this point in the paper.
> - In practice, we can thus embed multiple watermarks in the same image, and watermark only one object. However, the user might not deliberately place several watermarks, but only a single one on its whole image . Due to modifications, splicing, or inpainting, the image can end up with multiple watermarked regions, or one watermarked region and the rest not watermarked: this is more the motivation of the paper.
>
> > 2. Could you clarify how the mask for the watermark's position is inputted into WAM in Figure 7 of the Appendix? Additionally, could you explain why this specific position was chosen for the localization experiment?
>
> - Please refer to Section 5 where this is explained. The answer is similar to that of question 1: We watermark the entire image and then consider the splice between the original and watermarked images, retaining only the wm pixels whose positions are determined by the mask. This approach is similar to what is used in training and is depicted in Figure 2, with the corresponding equation in the augmentation paragraph of Section 4.2.  We will clarify this in the paper.
> - Regarding the specifics of the mask, this is described in Section 5.4.  We use centered watermarked rectangles of various sizes within the images to ensure a well-controlled experiment. This ensures that after the upper left crop, the same proportion of the resulting image remains watermarked. However, WAM can detect watermark areas of any shape (Figure 6 in the appendix shows the masks used in training, and WAM achieves an average mIoU of 95%).
>
>
> > 3. In lines 226-232, if I am not mistaken, this method appears to be identical to the method proposed by Bui et al., rather than merely 'a similar approach'. Please provide proper credit.
>
> - We do not cite the scaling as a contribution and will ensure this is more clearly highlighted in the paper.
> - However on a side note, the method by Bui et al. is not exactly the same, as they predict the watermarked image and then rescale, whereas we predict a distortion that is rescaled.
>
> > 4. Regarding the claim in lines 304-305, it is unclear why applying JND would make training easier compared to using perceptual loss. Could the authors provide evidence or reasoning to support this claim? 'Applying JND only in the second training phase makes training easier than previous methods relying on “perceptual” or “contradictory” losses.'
>
> - We do not claim that using JND is universally superior, but in our experiments, it proved easier to fine-tune with JND than with perceptual losses. Specifically, LPIPS did not perform visually as well (i.e. more visible) at a fixed PSNR compared to JND.
> - The rationale is that our 1M parameter embedder benefits from having the visual system provided via JND, eliminating the need for the relatively small network to learn it by itself and focus and message transmission. This simplifies training by reducing constraints.
> > 5. I believe the reason WAM outperforms the baselines in terms of bit accuracy for splicing is simply because WAM incorporates splicing during training, whereas the other models do not. If the other models were to apply splicing as an augmentation or attack during training, would WAM still outperform them in terms of bit accuracy?
>
> - Our embedder does not have a unique feature specifically for splicing; we agree that robustness to splicing comes from incorporating masks during training. We consider training with randomly generated masks a novel contribution of WAM.
> - The reviewer suggests that using the same training pipeline with masks, but without pixel-level detection, could still achieve robustness to splicing. We agree. However this is not done by others and it would be less informative, as it wouldn't identify which parts of the image are watermarked.
> - Note that WAM's pixel-level detection also enhances robustness against inpainting (as demonstrated in Table 2). Figure 10 in the appendix further illustrates the benefit of identifying inpainted pixels.

---

> > ### Author Response · Authors · 2024-11-19
> >
> > > 6. Typically, the TPR is reported at a fixed FPR, as seen in Tree Ring. Is the approach in this paper, reporting TPR at varying FPRs, fair and appropriate?
> >
> > - See ligne 409 in section 4.3: this is what we do: we show results at a fixed **theoretical** FPR. By stating “reporting TPR at varying FPRs”, does the reviewer mean that the empirical FPR slightly differs from the theoretical one? Yes. Another way to report the result would have been to fix the empirical FPR  by thresholding our detection scores. This is what we do for Tree ring in Table 6 of appendix.
> >
> > > 7. I am not entirely convinced of the necessity of embedding multiple messages separately into a single image. If the embedding capacity is large enough (e.g., 100 bits), users could already encode several different messages within a single bit stream. Could the authors provide some practical scenarios where embedding multiple separate messages would be particularly beneficial?
> >
> > - Please refer to Line 189 in Section 3 for more details. Embedding multiple messages separately in a single image offers several practical benefits:
> >
> > - **Robustness Against Splicing:** It enhances robustness against attacks involving the splicing of several watermarked images, which can compromise decoding in traditional watermarking schemes in real-world scenarios.
> > - **Active Object Detection:** It supports active object detection, where the goal is to track watermarked objects within an image.
> > - **Identification of Multiple AI Tools:** It allows for the identification and tracking of multiple watermarked AI tools used within a single image.
> >
> >
> > > 8. What happens if the position masks of multiple watermarks overlap? Additionally, for the watermark to be successfully extracted, what is the minimum proportion the watermark mask should cover relative to the overall image size? How many watermarks can be embedded in a single image at most?
> >
> > - See figure 4, which exactly answers the question of what proportion is necessary to detect/decode watermarks. It shows that starting from 5% of the image, one can decode a 32bit message with 90% accuracy.
> > The number of messages is not limited as long as it fits the image: we show in figure 5 that it works until 5 messages of 10% each, but it could be more (it works for 5 while we use a maximum of 3 different messages in the second phase of training, so it does generalize to more watermarks).
> > - We demonstrate that, on average across 100 images, WAM can still detect the presence of a watermark even if two watermarks overlap entirely on the whole image with different messages. We appreciate the reviewer for raising this point, as it highlights the advantage of having decoding separated from detection.
> >
> >
> > |                        | One Watermark | Two Overlapping Watermarks |
> > |--------------------|---------------|----------------------------|
> > | Detection Accuracy | 100%          | 100%                        |
> >
> > - However, WAM will not be able to decode both messages if one watermark is directly on top of the other. In practice, if the detector and encoder are managed by the same person, they can verify that the image is not already watermarked by WAM before adding a second watermark.
> >
> > > 9. A small typo on line 407, 'hide messages or arbitrary length' should be 'hide messages of arbitrary length'?
> >
> > Thank you
> >
> > > 10. Will the source code and pretrained model be released for the community?
> >
> > See ligne 523 of the conclusion. We commit to open source code and models.

---

> > > ### Author Response · Authors · 2024-11-19
> > >
> > > > 11. Is WAM robust to the new type of regeneration attacks [1], like diffusion-based regeneration?
> > >
> > > - We do demonstrate WAM's robustness against heavy inpainting, as shown in Table 2 and Figure 10 in the appendix. WAM is not only robust to such modifications but can also localize precisely what subpart has been edited.
> > > - Regarding full-image purification via Stable Diffusion, WAM shares the same limitations as other methods: we agree that this is a very important research topic, but this is beyond our current scope.
> > > - We still provide a new comparison here, focusing on decoding 32 bits, using the DiffPure method and implementation from [1]. We observe that WAM shows comparable robustness, and breaks when the purification is heavy. We will add this to the paper. (CIN and MBRS were added as asked by reviewer oRgo; see the corresponding answer for details)
> > >
> > > | Metric                | HiDDeN | WAM  | DWTDCT | SSL  | FNNS | TrustMark | MBRS | CIN  | WAM  |
> > > |-----------------------|--------|------|--------|------|------|-----------|------|------|------|
> > > | DiffPure_0.001 (PSNR 30.1) | 88.6   | 100  | 80.4   | 99.2 | 97.9 | 100.0     | 100.0| 100.0| 100.0|
> > > | DiffPure_005 (PSNR 28.4)   | 65.2   | 99.4 | 48.6   | 63.0 | 64.1 | 98.3      | 99.5 | 100.0| 99.4 |
> > > | DiffPure_01 (PSNR 27.0)    | 56.2   | 71.3 | 49.0   | 54.2 | 53.0 | 74.9      | 85.1 | 82.3 | 71.3 |
> > > | DiffPure_02 (PSNR 24.9)    | 51.6   | 49.1 | 49.5   | 52.0 | 50.1 | 54.3      | 64.8 | 61.9 | 49.1 |
> > >
> > > **Thank you**
> > >
> > > We thank the reviewer for their questions. We are surprised by the grade 2 to the “contribution” section: WAM offers three additional functionalities: localization, the ability to hide multiple 32-bit messages, and detection separate from decoding. If the reviewer is satisfied by our answers, we would be very grateful if they can consider raising their score.

---

> > > > ### Comment · Reviewer_UTj6 · 2024-11-22
> > > > **Question about how to achieve multiple watermarking**
> > > >
> > > > Authors mentioned: This process is detailed in Section 5.5: "This is done by feeding the image several times to WAM’s embedder, then pasting the different watermarked areas onto the original image."
> > > >
> > > > I would like to confirm if my understanding is correct.
> > > >
> > > > Does the embedder encode the entire image in every forward? If so, then the user needs to manually copy the desired watermarked region from this single forward according to a mask. This process will repeat n times, where n is equal to the number of objects or regions that the user wants to add watermarks. Is there anything wrong with my understanding?

---

> > > > ### Comment · Reviewer_UTj6 · 2024-11-22
> > > > **Question about the necessity of embedding multiple messages separately**
> > > >
> > > > The authors mentioned three benefits: (1) Robustness Against Splicing, (2) Active Object Detection, and (3) Identification of Multiple AI Tools.
> > > >
> > > > 1) It is indeed beneficial against splicing, but the methods trained with crop/cropout/dropout noise layers are also robust to splice already. I am still not sure why it is necessary for multiple separate messages?
> > > >
> > > > 2) For active object detection, could authors provide a practical scenario/example that this function is useful/important?
> > > >
> > > > 3) Could the authors clarify more in detail how separate messages can help identify and track multiple watermarked AI tools within a single image?

---

> ### Author Response · Authors · 2024-11-22
>
> We thank the reviewer for their response.
>
> > Does the embedder encode the entire image in every forward? If so, then the user needs to manually copy the desired watermarked region from this single forward according to a mask. This process will repeat n times, where n is equal to the number of objects or regions that the user wants to add watermarks. Is there anything wrong with my understanding?
>
> - Your understanding is correct. The embedder encodes the entire image in each forward pass, and the user then uses a mask to replace the original part of the image by its watermarked version (see equation in line 247). This process is repeated for each object or region the user wishes to watermark. Note that a segmentation model can be employed to create the mask for the specific object or region.
> -  We would like to emphasize that our primary motivation for WAM is watermark an entire image, which may then undergo various transformations such as inpainting or background changes. This can result in only a portion of the image retaining the watermark (or several watermarks in the same image). While it is indeed possible to deliberately watermark different objects within the same image, this is a secondary consideration. We will make sure to clarify this aspect.
>
> About multiple watermarks:
>
> > 1. It is indeed beneficial against splicing, but the methods trained with crop/cropout/dropout noise layers are also robust to splice already. I am still not sure why it is necessary for multiple separate messages?
>
> - If a GenAI provider applies a unique watermark message for each user, and the images are later spliced, there will be different messages embedded in different parts of the resulting image. WAM can successfully extract each distinct message. In contrast, methods that extract only a single message would inevitably fail in this scenario, even if they are trained with crop/cropout/dropout noise layers as suggested by the reviewer as they decode a single message per image, they will not be able to extract several messages.
>
> > 2. For active object detection, could authors provide a practical scenario/example that this function is useful/important?
>
> - A practical scenario where active object detection is useful is in tracking the usage of specific objects across different posts (see [1]), such as in memes or other derivative works . By embedding a watermark in an object within an image, you can monitor how that object is reused elsewhere. This is particularly valuable for copy detection, as it allows you to identify and trace the spread of the object across various contexts. Additionally, a watermarked object can deliberately serve as a visual hashtag, making it easily detectable by the extractor and facilitating the organization and retrieval of content related to that object.
>
> > 3. Could the authors clarify more in detail how separate messages can help identify and track multiple watermarked AI tools within a single image?
>
> - For instance, if you own a tool that can modify images in various ways, you can add a unique watermark message to each modification. This means that if one user changes someone's hair color using your tool, the modified hair color can be watermarked with a specific message (e.g., user ID or type of transformation ID). Later, if another user changes the background of the same image using a different feature of your tool, the new background can be watermarked with a distinct message. So by embedding these separate messages, it becomes possible to track the history of modifications made to the image.
>
> Overall, we think that it is just a more versatile way of seeing image watermarking: it also keeps all the natural properties of other models. We remain open to further clarifications.
>
> [1] Vishal Asnani, Abhinav Kumar, Suya You, and Xiaoming Liu. Probed: proactive object detection
> wrapper. Advances in Neural Information Processing Systems, 36, 2024.

---

> > ### Comment · Reviewer_UTj6 · 2024-11-22
> >
> > Thanks for the authors' patient explanation and insightful discussion. It now makes sense to me. Most of my concerns are addressed. I am pleased to raise the score.

---

> > > ### Author Response · Authors · 2024-11-22
> > > **Thank you!**
> > >
> > > We thank the reviewer for their valuable feedbacks. We will clarify the paper accordingly.

---

### Official Review · Reviewer_pZCp · 2024-10-28

**Soundness:** 3
**Presentation:** 3
**Contribution:** 4
**Rating:** 8
**Confidence:** 3

**Summary:**

The paper generalizes for multi-source images and achieves image locality preservation without a priori knowledge. The most innovative is to define watermarking as a segmentation task. Other contributions have been proposed in state-of-the-art papers, not for the first time in this thesis, and are not innovative enough.

**Strengths:**

The paper introduces the Watermark Anything Model (WAM), a deep learning-based approach for embedding and extracting localized image watermarks, which excels in efficiently embedding and locating watermarks within small areas, demonstrating resilience against image tampering and splicing. It can extract distinct 32-bit messages from multiple regions, each less than 10% of the image area, with minimal bit error rates. Trained at low resolutions and further optimized for perception invisibility and multiple watermarks, WAM competes with state-of-the-art methods in terms of imperceptibility and robustness, offering new possibilities and enhanced practicality for image watermarking technology.

**Weaknesses:**

1. Insufficient innovation, little change in the existing image watermarking framework, no analysis for specific modules, still using the traditional training architecture, just increase the template after random clipping, to achieve the application of the scene changes.
2.Many existing work has been involved in local image watermarking, known as multi-source image watermarking [1]. The arbitrary image watermarking framework proposed in this paper though does not require prior knowledge and can be protected against any image. But only the major part of the image in a real scene has a role to play and there is no need to add watermarking information for any pixel.

[1] Wang G, Ma Z, Liu C, et al. MuST: Robust Image Watermarking for Multi-Source Tracing[C]//Proceedings of the AAAI Conference on Artificial Intelligence. 2024, 38(6): 5364-5371.

**Questions:**

1. The paper compares EditGuard's work, but does not compare the work that is very similar to this paper [1], please explain the difference between the two;
2. How to guarantee the imperceptibility of the framework, and did not see the loss module in the mature watermarking framework, such as the MSE loss between the embedded image x_m and x, the loss of discriminative networks, etc.

---

> ### Author Response · Authors · 2024-11-19
>
> > Weakness 1: Insufficient innovation, little change in the existing image watermarking framework, no analysis for specific modules, still using the traditional training architecture, just increase the template after random clipping, to achieve the application of the scene changes.
>
> - WAM is the first work to approach watermarking as a segmentation task, which significantly differs from traditional architectures. It offers three additional functionalities: localization, the ability to hide multiple 32-bit messages, and detection separate from decoding.
> - We are unclear about the reviewer's comment, "just increase the template after random clipping, to achieve the application of the scene changes." Could the reviewer please clarify?
> - If the suggestion is to simply add masks as augmentation without performing segmentation/detection, we acknowledge that this might result in a model robust to splicing. However, it would not allow us to identify which parts of the image are watermarked after modifications, which is WAM's primary motivation. Additionally, it would not support embedding multiple watermarks, which is another key innovation of our approach. Moreover, an innovation in WAM is specifically this training with random masks.
>
> > Weakness 2: Many existing work has been involved in local image watermarking, known as multi-source image watermarking [1]. The arbitrary image watermarking framework proposed in this paper does not require prior knowledge and can be protected against any image. But only the major part of the image in a real scene has a role to play and there is no need to add watermarking information for any pixel.
>
> - The statement that "the major part of the image in a real scene has a role to play and there is no need to add watermarking information for any pixel" is subjective and open to discussion. It's important to note that a user might not intentionally choose to watermark only the background. In scenarios where the entire image is watermarked but only the background remains after inpainting the foreground by someone else, WAM can segment and identify the watermarked areas (see Table 2 and Figure 10 in the appendix). Simply providing a yes or no answer for the entire image would be insufficient.
>
> > 1 The paper compares EditGuard's work, but does not compare the work that is very similar to this paper [1], please explain the difference between the two;
>
> - Thank you for highlighting the reference to MuST [1], we will cite it. Unlike MuST, which decodes different parts of the image independently, WAM uses a single extractor that automatically performs segmentation. This allows for a watermarked background, which is specifically why we opted to predict masks rather than boxes, because it offers greater versatility
> - Unfortunately, MuST's weights are not available, and their GitHub repository does not provide instructions for reproducing the model (as noted in issues raised by others), so we cannot offer a direct comparison.
> - But considering the results presented in the paper, MuST is tailored for specific applications and struggles with scenarios like a 10% crop, where it results in random bit accuracy. MuST loses robustness even at an 80% crop.
>
> > 2) How to guarantee the imperceptibility of the framework, and did not see the loss module in the mature watermarking framework, such as the MSE loss between the embedded image x_m and x, the loss of discriminative networks, etc.
> This is detailed in Section 4.3 and illustrated in Figure 3.
>
> - We do not use perceptual losses (e.g., LPIPS or MSE) or a discriminator, as these can introduce instability due to conflicting signals. Instead, WAM's training is divided into two phases:
>
> **First Phase:** Training is conducted without any perceptual loss/constraint, using only the tanh function to upper bound the watermark distortion:
>
> $$
>    x_{\text{wm}} = x + \alpha_1 \times \tanh (\delta_{\theta}(x,m))
> $$
>
>    We continue training until achieving good detection and decoding accuracy, which serves as an initialization for the second phase.
>
> **Second Phase:** Training proceeds similarly (using the same augmentations/masks for robustness/detection), but now the distortion is modulated by the Just Noticeable Difference (JND):
>
> $$
>    x_{\text{wm}} = x + \alpha_{\text{JND}} \cdot \mathrm{JND}(x) \odot \tanh(\delta_{\theta}(x,m))
> $$
>
> We do not state that it is optimal, but experimentally it eases things as the small embedder does not have to learn the visual system.

---

> > ### Comment · Reviewer_pZCp · 2024-11-21
> >
> > Regarding my previous comment "just increase the template after random clipping, to achieve the application of the scene changes", I would like to provide further explanation. First of all, I recognize the innovation of your work in summarizing the watermarking task as a segmentation task, which provides a new approach to solving the watermarking problem. However, as for the subsequent innovation points, such as the two-stage training, they have been involved in the existing research and are not first proposed in this paper. Nevertheless, overall, your idea of transforming the watermarking task into a segmentation task alone is leading. Through reviewing the details of the paper, referring to your responses to other reviewers, and the experimental results, I think your work is generally of high quality.
> > Specifically, in the experimental part, your work shows a high standard. For example, in the experiment on the ability to handle multiple watermarks, you have conducted a detailed evaluation of the model's ability to extract multiple watermarks in different training stages, and the results clearly demonstrate the performance of the model, which is worthy of recognition. At the same time, when compared with other methods, the advantages of the proposed method in this paper can also be highlighted. For instance, when comparing the watermark localization ability with EditGuard, it is clearly shown that WAM has better performance in different situations.
> > However, there are still some aspects that can be further optimized. For the model architecture, although the current architecture can already achieve basic functions, considering the development needs of watermarking technology, exploring how to build a larger-scale model while maintaining efficient processing capabilities may further improve performance. For example, it may have more advantages in handling watermark embedding and extraction in more complex scenarios. In terms of watermark invisibility, although methods such as JND weighting have been adopted, there is still room for improvement. You can further study models that are more in line with the characteristics of the human visual system or perform more effective regularization on the watermark signal during the training process to reduce the appearance of visible watermarks, especially in complex images or specific regions (such as high-contrast areas). I hope you can continue to improve your work according to these suggestions, and I look forward to seeing more excellent results.

---

> > > ### Author Response · Authors · 2024-11-21
> > >
> > > We thank the reviewer for their response.
> > >
> > > Given your comments that acknowledges the novelty and high quality of the experiments, do you consider the current body of work worth publishing? If so, would you consider raising your score?

---

### Official Review · Reviewer_oRgo · 2024-11-02

**Soundness:** 3
**Presentation:** 2
**Contribution:** 3
**Rating:** 8
**Confidence:** 4

**Summary:**

This paper addresses the limitation of current image watermarking methods in handling small watermarked areas. The authors redefine the watermarking task as a segmentation problem, decomposing it into localization and detection tasks. They adopt a two-stage training strategy to separately achieve robustness and imperceptibility. By incorporating localization capabilities, the authors enable embedding multiple watermarks within a single image. Extensive experiments demonstrate the broad applicability of their method.

**Strengths:**

1. The paper introduces an innovative approach by transforming the watermarking task into a segmentation task, enabling the embedding of multiple watermarks at different locations in an image and supporting watermarking of small areas.
2. The proposed method shows broad applicability and can handle high-resolution images and images of various sizes.
3. The design motivations are well-justified, and the implementation details are clear. For improving visual quality, the authors opted for the JND approach instead of common "perceptual" or "contradictory" losses. Extensive experiments validate the effectiveness of using JND for watermarking.

**Weaknesses:**

1. Although the authors discuss different model architectures, the overall parameter size remains significantly larger compared to prior models such as HiDDeN (454.4K), FIN (737.8K) [2], and CIN (36.01M) [3]. While the authors highlight that their encoder (1.1M parameters) is relatively compact, it is still substantially larger compared to earlier models like HiDDeN (188.93K) and FIN (747.80K). This large parameter size may limit the method's applicability in resource-constrained environments.
2. The comparative experiments are not comprehensive. The authors do not consider more advanced watermarking models, such as MBRS, FIN, and CIN, which significantly outperform HiDDeN.
3. It would be helpful for the authors to provide a relationship graph between the extractor’s parameter size and performance, to justify the necessity of the large parameter size for detection and decoding tasks.

Overall, I am optimistic about the paper's method and the problem it addresses. I am particularly intrigued by the authors' segmentation-based approach for embedding multiple watermarks within a single image. My final score will depend on the authors' rebuttal.


[1] Fang H, Qiu Y, Chen K, et al. Flow-based robust watermarking with invertible noise layer for black-box distortions[C]//Proceedings of the AAAI conference on artificial intelligence. 2023, 37(4): 5054-5061.

[2] Ma R, Guo M, Hou Y, et al. Towards blind watermarking: Combining invertible and non-invertible mechanisms[C]//Proceedings of the 30th ACM International Conference on Multimedia. 2022: 1532-1542.

**Questions:**

See weaknesses.

---

> ### Author Response · Authors · 2024-11-19
>
> We thank the reviewer for their valuable feedback and time spent on our paper.
>
> > 1) Although the authors discuss different model architectures, the overall parameter size remains significantly larger compared to prior models such as HiDDeN (454.4K), FIN (737.8K) [2], and CIN (36.01M) [3]. While the authors highlight that their encoder (1.1M parameters) is relatively compact, it is still substantially larger compared to earlier models like HiDDeN (188.93K) and FIN (747.80K). This large parameter size may limit the method's applicability in resource-constrained environments.
>
> - We agree that smaller models enhance applicability for memory-constrained envs, but a larger parameter size does not necessarily equate to slower performance. Our method consistently embeds or detects the watermark at a low resolution (256x256). This downscaling and subsequent upscaling approach, not used in HiDDeN, MBRS, or SSL, ensures that WAM operates with a constant computational requirement regardless of image size.
> - Our watermark embedder indeed consists of 1M parameters, which is larger than HiDDeN. However, this increase in size does not inherently lead to slower performance. Below is a comparison of the WAM embedder's speed to HiDDeN's in seconds on a 32GB V100 GPU (with 16 CPUs, Intel(R) Xeon(R) Gold 6230 CPU @ 2.10GHz):
>
> | Method         | HiDDeN | WAM  | DWTDTC | SSL   | FNNS  | TrustMark | MBRS  | CIN   |
> |----------------|--------|------|--------|-------|-------|-----------|-------|-------|
> | Encoding Time  | 0.070  | 0.040| 0.041  | 0.228 | 0.508 | 0.026     | 0.052 | 0.124 |
> | Decoding Time  | 0.026  | 0.032| 0.033  | 0.017 | 0.024 | 0.022     | 0.028 | 0.074 |
>
>
> We appreciate the reviewer highlighting this point and will include a discussion in the paper. Additionally, as mentioned in the general comment, we will add an ablation study on using smaller embedders or extractors.
>
>
> > 2) The comparative experiments are not comprehensive. The authors do not consider more advanced watermarking models, such as MBRS, FIN, and CIN, which significantly outperform HiDDeN.
>
> - We chose to compare the methods present in the paper because they constitute a solid baseline. Our goal was not to demonstrate that WAM is more robust than all methods in every setting, but rather to show that it performs well for classical watermarking while introducing new capabilities.
>
> - We argue that MBRS, FIN, and CIN are not robust to crops, indicating a different operating point. Our focus is on decoding the image even from small watermarked areas.
>
> We have added comparisons to CIN and MBRS*. Specifically:
>
> - MBRS has a 256-bit payload. Like other methods, we encode 16 bits for detection and 32 bits for decoding. We encode this 16+32 = 48-bit message 5 times, occupying 240 bits. At decoding time, we average the outputs of the decoder every 48 bits (soft majority vote) and output the decoded 48-bit message.
> - CIN, on the other hand, only handles 30 bits. We show the bit accuracy for the first 30 bits of the message for CIN and do not perform detection (a favorable setup for CIN).
>
> - For both methods, we scale the distortion to achieve a similar PSNR as all models (see Table 1). We present the bit accuracy on the validation set of COCO, using the exact same setup as in Table 2:
>
>
> | Method | None          | Geometric     | ValueMetric   | Splicing      | PSNR/SSIM/LPIPS |
> |--------|---------------|---------------|---------------|---------------|-----------------|
> | MBRS   | 100.0 | 50.4  | 99.8 | 65.31  | 38.8/0.99/0.07 |
> | CIN    | 100.0 | 50.3  | 100.0 | 96.47 | 38.3/0.99/0.08 |
> | WAM    | 100.0 | 91.8   | 100.0 | 95.3 | 38.3/0.99/0.04 |
>
> - We observe that both methods do not handle any geometric transformations (they actually only manage crop and then padding, not crop and resize, which is not a realistic setup). This is due to their different operating points, which place more emphasis on robustness against value metric transformations.
>
> > 3) It would be helpful for the authors to provide a relationship graph between the extractor’s parameter size and performance, to justify the necessity of the large parameter size for detection and decoding tasks.
>
> - Our preliminary ablations indicated that a larger extractor is necessary to enhance pixel-level detection and decoding. We agree that including an ablation study on this topic could be valuable, as smaller models might benefit the community even if they perform slightly worse. As mentioned in the general comment, we have launched the training and will include the results in the paper.
>
> > Overall, I am optimistic about the paper's method and the problem it addresses. […]. My final score will depend on the authors' rebuttal.
>
> - We thank the reviewer for the positive feedback and are glad to hear of your interest. We remain open for discussion during the rebuttal session.
>
> *we had to revise the table in the comment because of a typo

---

> > ### Comment · Reviewer_oRgo · 2024-11-19
> >
> > Firstly, I sincerely thank the authors for their detailed rebuttal, which has effectively addressed several of my earlier concerns. However, I have some new points that I would like to raise:
> >
> > 1. **Regarding the first question**, I find the authors' perspective reasonable—larger parameters do not necessarily imply greater computational complexity. However, I still have a few concerns:
> >    - Why didn’t the authors provide a more objective and universal evaluation metric for computational complexity, such as FLOPs? FLOPs are independent of experimental environments, which can vary significantly among different users.
> >    - When I referred to "resource-constrained environments," I was not only highlighting limited computational resources but also restricted storage resources. The storage requirements for deploying a model are closely tied to its parameter count. On this front, I believe the proposed model still has room for improvement.
> >
> >    While the second concern is more of a suggestion for future work, I strongly recommend that the authors include FLOPs data to address the first point.
> >
> > 2. **Regarding the second question**, I appreciate the authors’ inclusion of comparative experiments with MBRS and CIN based on my earlier suggestion. However, I have some questions about the experimental setup:
> >    - Both MBRS and CIN offer public implementations supporting variable message lengths. Why did the authors embed a length of 256 bits in MBRS instead of directly using 48 bits? CIN explicitly supports 48-bit embeddings, so I recommend the authors revisit CIN's public codebase to verify this.
> >    - What Noise Layer settings were employed during the evaluation of MBRS and CIN? The distortion types in the Noise Layer substantially impact the robustness of these models. Did the authors use the same Noise Layer settings specified in the original papers to ensure experimental fairness, particularly regarding geometric transformations?

---

> ### Author Response · Authors · 2024-11-19
>
> We thank the reviewer for their prompt answer
>
> Regarding 1)
>
> - 22 Giga flops for HiDDeN VS 42 Giga Flops for our VAE embedder when evaluating both on 256x256 images. We reported speed because it was asked by reviewer n1mi. We agree that both are important metrics; flops will be added in the table.
> - We agree about the parameter counts, which would be in favor of HiDDeN. But we would argue that a 1.1M parameter is still doable in most scenarios. For instance, with int8 quantization, it would fit in 1MB. (Note that this is less the case for the extractor, but this is less of an issue since most of the time detection is done on the server side.)
>
> Regarding 2)
>
> - We use the public models available, that don't have different capacities. We did not have time to retrain models from scratch for the rebuttal. For MBRS, we chose the model that was at the same resolution (256x256), and for CIN, we chose the only one that was available.
> - For the noise layers, these models don't use any augmentations that change the geometry of the image (the crop they use is not a crop, but a black mask pasted on top of part of the image - see https://github.com/rmpku/CIN/blob/main/codes/models/modules/Noise_option/crop.py), which may explain the performance they show. For our evaluations, we use geometric augmentations as defined by the torchvision library (for instance, the crop we evaluate on is a "true" crop). We agree that its not apple to apple comparaison, but we believe that our transformations are more realistic to evaluate robustness against real-world application. Moreover, in our experience, as soon as you add crops/perspective or any geometric changes, training is much harder. We could ask the authors of the paper if they have tried  training with it.

---

> > ### Comment · Reviewer_oRgo · 2024-11-26
> > **My decisions and some further suggestions**
> >
> > I am positive about the problem the paper aims to solve (i.e., image watermarking methods not being tailored to handle small watermarked areas) and the solution proposed by the authors, which redefines watermarking as a pixel-level segmentation task. This is also why I initially gave a positive score to the paper. After carefully reviewing the other reviewers' comments, I have decided to maintain my positive evaluation.
> >
> > However, I do have some suggestions regarding the authors’ rebuttal. Specifically, I believe the comparison experiments on geometric robustness (MBRS, CIN) and the conclusion "both methods do not handle any geometric transformation" should not be included in the main text now. As you mentioned, "we use the public models available," but the noise layer in these public models are not trained for the geometric distortions present in your experimental setup. Therefore, the experimental setup is incomplete, and the conclusion is not solid. If the paper is accepted, I hope you will have more time to refine this section and present more convincing results in the final published version.

---

> > > ### Author Response · Authors · 2024-11-26
> > >
> > > We appreciate the reviewer's engagement throughout the rebuttal process.
> > >
> > > We will ensure that the limitations of our comparisons are clearly highlighted. We understand the reviewer's point that MBRS and CIN were not trained with these specific noise layers which will be emphasised. We will do our best, and are open to suggestions from the reviewer on how to make the experimental set-up more complete
> > >
> > > Our objective was to demonstrate comparable robustness to other methods while **offering additional properties**, rather than claiming superiority in all aspects. Given that each model is optimised for different operating points concerning perceptibility, capacity, and robustness, making absolute quantitative comparisons is challenging. We have tried to compare to many OS models while making our comparisons as relevant as possible by incorporating a comprehensive set of common image augmentations that are realistic in the internet/social network sharing setting that we care about (in particular, it's important to note that we do not omit transformations for which other models are trained to be robust against, while WAM was not).
> > >
> > >
> > >
> > > We thank again the reviewer for their positive feedback.

---

### Official Review · Reviewer_n1mi · 2024-11-04

**Soundness:** 2
**Presentation:** 3
**Contribution:** 2
**Rating:** 5
**Confidence:** 4

**Summary:**

The paper presents a novel framework, WAM, for localized image watermarking that embeds and extracts watermarks from specific regions within an image. WAM’s key innovation is treating watermarking as a segmentation task, which enables the model to identify and decode watermarked regions even when only small parts of an image are modified. The model operates in two phases: initial training for robustness and a fine-tuning stage to ensure imperceptibility and support for multiple watermarks.

**Strengths:**

1. WAM redefines watermarking as a pixel-level segmentation task, allowing for precise localization and recovery of watermarks within small image regions. This advancement expands watermarking capabilities beyond global, image-wide methods.
2. WAM’s architecture combines an encoder-decoder structure and clustering techniques, making it well-suited for segmentation. The two-stage training, which introduces imperceptibility and multi-message support, adds flexibility and enhances usability in high-resolution and high-precision tasks.

**Weaknesses:**

1. Although the paper compares WAM with several baseline watermarking models, including advanced steganographic methods could provide a more comprehensive assessment of WAM's relative strengths and limitations. Additionally, embedding and extracting multiple watermarks in different regions of an image is not a unique concept, as it has been previously introduced in methods such as [a] and [b].
2. The need for pixel-level detection across high-resolution images could make WAM computationally intensive, especially in scenarios requiring real-time analysis. Further evaluation of computational costs would help contextualize its practical limitations.
3. The paper acknowledges WAM’s limited payload capacity (32 bits per region). While suitable for smaller messages, the model may face challenges in applications requiring larger payloads. Future work could explore optimizing the architecture to balance capacity and robustness.
4. The reliance on DBSCAN parameters for multi-watermark clustering introduces an element of complexity. Parameter tuning may be required across different datasets or conditions, potentially complicating deployment.

[a] Pan, Wenwen, Yanling Yin, Xinchao Wang, Yongcheng Jing, and Mingli Song. "Seek-and-hide: adversarial steganography via deep reinforcement learning." IEEE Transactions on Pattern Analysis and Machine Intelligence 44, no. 11 (2021): 7871-7884.
[b] Meng, Ruohan, Qi Cui, Zhili Zhoul, Chengsheng Yuan, and Xingming Sun. "A Novel Steganography Algorithm Based on Instance Segmentation." Computers, Materials & Continua 63, no. 1 (2020).

**Questions:**

Please clarify the inovitive of WAM accoding to the weakness.

---

> ### Author Response · Authors · 2024-11-19
>
> We thank the reviewer for their valuable feedback and time spent on our paper.
>
> > 1) Although the paper compares WAM with several baseline watermarking models, including advanced steganographic methods could provide a more comprehensive assessment of WAM's relative strengths and limitations. Additionally, embedding and extracting multiple watermarks in different regions of an image is not a unique concept, as it has been previously introduced in methods such as [a] and [b].
>
> - The official terminology established by Ross Anderson and Birgit Pfitzmann at the Information Hiding conference in the 90s distinguishes between steganography (hiding messages undetectably) and digital watermarking (hiding messages robustly). This distinction is widely accepted in the literature, as noted in the seminal book by Ingemar Cox et al. The reviewer seems to be conflating steganography with digital watermarking. Based on their titles, references [a] and [b] are steganography papers.
> - Reference [a] involves hiding an image within another, albeit locally, which is a fundamentally different approach.
> - We appreciate the reviewer mentioning reference [b], but it does not offer watermark detection and is not shown to have the capability to hide multiple watermarks.
> - We have added comparisons* to CIN [1] and MBRS [2], as suggested by reviewer oRgo (see our response to the reviewer for implementation details). However, these methods do not handle localization or multiple watermarks. Below is the decoding accuracy when a 32-bit message is hidden, obtained for the same columns as Tables 1 and 2:
>
>
> | Method | None          | Geometric     | ValueMetric   | Splicing      | PSNR/SSIM/LPIPS |
> |--------|---------------|---------------|---------------|---------------|-----------------|
> | MBRS   | 100.0 | 50.4  | 99.8 | 65.31  | 38.8/0.99/0.07 |
> | CIN    | 100.0 | 50.3  | 100.0 | 96.47 | 38.3/0.99/0.08 |
> | WAM    | 100.0 | 91.8   | 100.0 | 95.3 | 38.3/0.99/0.04 |
>
> - We can see that both methods do not handle any geometric transformation, because they have different operating points (larger capacity, more emphasis on robustness against value metric transformations).
>
> > 2) The need for pixel-level detection across high-resolution images could make WAM computationally intensive, especially in scenarios requiring real-time analysis. Further evaluation of computational costs would help contextualize its practical limitations.
>
> - This is discussed in section 6 (Limitations). WAM consistently embeds or detects the watermark at a low resolution (256x256). This downscaling and subsequent upscaling approach, not used in HiDDeN, MBRS, or SSL, ensures that WAM operates at a constant computational requirement regardless of the image size.
>
> - Our watermark embedder consists of 1M parameters, larger than the others. However, this increase in size does not necessarily translate to slower performance; it primarily requires more memory.
> - The pixel-level detection capability is indeed why we found that increasing the size of WAM's extractor enhances decoding and detection performance. However, similarly to the encoder, the number of parameters does not directly correlate with speed
>
> We provide a comparison of the WAM embedder's speed (in seconds) to others on a one 32GB-V100 GPU (with 16 cpus Intel(R) Xeon(R) Gold 6230 CPU @ 2.10GHz):
>
>
> | Method         | HiDDeN | WAM | DWTDTC | SSL | FNNS | TrustMark | MBRS | CIN |
> |----------------|-------------|------------------------|--------|---------|----------|-------------|----------|----------|
> | encoding time  | 0.070       | 0.040                  | 0.041  | 0.228   | 0.508    | 0.026       | 0.052    | 0.124    |
> | decoding time   | 0.026       | 0.032                  | 0.033  | 0.017   | 0.024    | 0.022       | 0.028    | 0.074    |
>
> However, a larger number of parameters does correlates with larger memory needs. As stated in the general comment, we will add an ablation on using smaller models.
>
> *we had revise the table in the response because of a typo in the original one.

---

> > ### Author Response · Authors · 2024-11-19
> >
> > > 3) The paper acknowledges WAM’s limited payload capacity (32 bits per region). While suitable for smaller messages, the model may face challenges in applications requiring larger payloads. Future work could explore optimizing the architecture to balance capacity and robustness.
> >
> > - We kindly invite the reviewer to review our general comment about that point. More specifically, this is the operating point we have chosen intentionally, but a different one with less robustness (especially to geometric transformations) could have been selected.
> >
> > **32 Bits is Sufficient:**
> > See limitations in Section 6. Unlike other methods that rely on decoding to verify the  presence of the watermark using some of the capacity, WAM separates detection from decoding, making 32 bits sufficient e.g.  for labeling each GenAI provider differently, and even if the watermarked zone is only 10% of the image.
> >
> > **How to Increase Capacity:**
> > Our focus is intentionally on 32 bits. However, for larger payloads:
> > - Extending training (e.g., to 800 epochs) could increase capacity. We are testing this and will include findings in the camera-ready version.
> > - Increasing the localization constraint from 10% to 20% of 256x256 or training on 512x512 resolution while maintaining a 10% target area could also increase capacity.
> > - Lighten the robustness constraints during training.
> >
> >
> > > 4) The reliance on DBSCAN parameters for multi-watermark clustering introduces an element of complexity. Parameter tuning may be required across different datasets or conditions, potentially complicating deployment.
> >
> > - We agree with the reviewer that reliance on DBSCAN parameters introduces complexity. However, our experiments indicate that DBSCAN is not overly sensitive to hyperparameter tuning (see Figure 10 in the appendix). While other clustering methods could be considered, DBSCAN is one of the methods with the fewest parameters, making it a practical choice.
> >
> > [1] Fang H, Qiu Y, Chen K, et al. Flow-based robust watermarking with invertible noise layer for black-box distortions[C]//Proceedings of the AAAI conference on artificial intelligence. 2023, 37(4): 5054-5061.
> > [2] Ma R, Guo M, Hou Y, et al. Towards blind watermarking: Combining invertible and non-invertible mechanisms[C]//Proceedings of the 30th ACM International Conference on Multimedia. 2022: 1532-1542.

---

> > > ### Author Response · Authors · 2024-11-27
> > >
> > > Dear Reviewer,
> > >
> > > We wanted to follow up to ensure that our responses to your comments were satisfactory.
> > > Currently, your score is below the acceptance threshold. We hope our rebuttal has addressed your concerns and that, if you believe Watermark Anything is worth publishing, you might consider raising it.
> > >
> > > Please feel free to reach out if you require any further clarifications. And thank you again for your time!

---

> > > > ### Author Response · Authors · 2024-12-02
> > > >
> > > > Dear Reviewer,
> > > >
> > > > Today is the final day for reviewers' comment posting. We would appreciate any updates you can provide.
> > > >
> > > > Thank you!
> > > >
> > > > The Authors

---

### Official Review · Reviewer_mbVM · 2024-11-05

**Soundness:** 2
**Presentation:** 2
**Contribution:** 2
**Rating:** 5
**Confidence:** 4

**Summary:**

This paper introduced a new approach to image watermarking called the Watermark Anything Model (WAM). It allowed identify and localize the watermarked regions rather than just determined if an image is watermarked.
WAM used a two-stage training process. In the first stage, based on the traditional LDM encoder-decoder architecture, the watermark messages were embedded into every single pixel of the image. Detection and Decoding were then measured by calculating the portion or weighted average of the detected watermarked pixels. The second stage applied the Just-Noticeable-Difference map to make the watermark more imperceptible.
 This paper introduced a novel approach that allowed the localization of the watermarked regions; it proposed a two-stage training process rather than traditional DNN-based method that balanced between robustness and imperceptibility; It allowed multiple distinct watermarks to be embedded into a single image.

**Strengths:**

Originality:
WAM aimed to localize the watermarked region within an image, which is a novel perspective in watermarking. On the other hand, WAM was able to handle multiple distinct watermarks within a single image, which was rarely addressed in prior work. Additionally, WAM has a two-stage training process, first stage trained robustness while the second stage trained imperceptibility,  which is different from the traditional DNN-based methods.

Quality:
The authors provided comprehensive experiments on standard datasets, which strengthened the validity of the results.

Clarity:
The paper was well-structured and clearly explained. The use of toy examples and tables enhanced the readability and understanding of the paper.

Significance:
 The paper introduced watermark localization and a two-stage training process, which could open up new possibilities in watermarking technology.

**Weaknesses:**

1.	Limited capacity
WAM’s capacity was limited to 32 bits, which may be insufficient for many real-world applications. On the other hand, the comparison with the baseline method were not entirely fair when they are designed for larger capacity.
2.	Although utilized the JND map to improve watermark imperceptibility, it was only comparable to, but not outstanding among, baseline methods. On the other hand, since WAM used a two-stage training process, first training for robustness and then for imperceptibility, it was unclear how well WAM balanced the trade-off between robustness and imperceptibility when training these aspects separately.
3.	 The paper lacked discussion of the computational complexity, which would be valuable for understanding its efficiency and scalability in real-world applications.

**Questions:**

1.	In the two-stage training process, robustness and imperceptibility were trained separately.  Could the authors clarify how they ensure that this separation does not lead to conflict? For instance, robustness training led to visible artifacts, while imperceptibility training weakened robustness. How is this balance maintained effectively?
2.	Could the authors discuss the trade-off between the bit capacity, robustness, imperceptibility? It is well-known that there is an inherent trade-off among these three dimensions in watermarking. In this paper, bit capacity was sacrificed, yet imperceptibility was not outstanding than the baseline, and robustness comparisons were limited to a specific scenario: localized watermarked regions.
3.	While WAM used a JNP map to enhance imperceptibility, its imperceptibility was comparable but not outstanding relative to baseline methods. Is there potential for further improvement in imperceptibility without compromising robustness?
4.	Given that the capacity of WAM was limited, its ability to support multiple distinct watermarks is constrained. As the amount of AIGC grows rapidly, do the authors have suggestions for extending WAM to support larger payload?

---

> ### Author Response · Authors · 2024-11-19
>
> We thank the reviewer for their valuable feedback and time spent on our paper.
>
> > 1) W 1 [...] the comparison with the baseline method were not entirely fair when they are designed for larger capacity, i.e. SSL’s default capacity is 2048 bits.
>
> - We believe there is a misunderstanding. SSL cannot hide 2048 bits; it only encodes 30 bits as stated in the SSL paper. While it could potentially encode more, doing so would compromise both perceptibility and robustness.
>
> > The paper lacked discussion of the computational complexity, which would be valuable for understanding its efficiency and scalability in real-world applications.
>
> - This is discussed in Section 6,  limitation paragraph. We agree that smaller models enhance applicability for memory-constrained envs, but a larger parameter size does not necessarily equate to slower performance. Our method consistently embeds or detects the watermark at a low resolution (256x256). This downscaling and subsequent upscaling approach, not used in HiDDeN, MBRS, or SSL, ensures that WAM operates with a constant computational requirement regardless of image size.
> - WAM's watermark embedder indeed consists of 1M parameters, which is larger than e.g. HiDDeN, and our extractor is significantly larger (price to pay to get pixel level detection). However, this increase in size does not inherently lead to slower performance. Below is a comparison of the WAM embedder's speed to others in seconds on a 32GB V100 GPU (with 16 CPUs, Intel(R) Xeon(R) Gold 6230 CPU @ 2.10GHz):
>
> | Method         | HiDDeN | WAM  | DWTDTC | SSL   | FNNS  | TrustMark | MBRS  | CIN   |
> |----------------|--------|------|--------|-------|-------|-----------|-------|-------|
> | Encoding Time  | 0.070  | 0.040| 0.041  | 0.228 | 0.508 | 0.026     | 0.052 | 0.124 |
> | Decoding Time  | 0.026  | 0.032| 0.033  | 0.017 | 0.024 | 0.022     | 0.028 | 0.074 |
>
> - We agree that this is an important point that we will clarify. Also as pointed out in the general comment, we will add an ablation for different extractor sizes.
>
> > 1)  In the two-stage training process, robustness and imperceptibility were trained separately. Could the authors clarify how they ensure that this separation does not lead to conflict? For instance, robustness training led to visible artifacts, while imperceptibility training weakened robustness. How is this balance maintained effectively?
> This is described in Section 4.3 and illustrated in Figure 3.
>
> This is described in Section 4.3 and illustrated in Figure 3.
>
> During the first training phase, the training is conducted without any perceptual loss/constraint, using only the tanh function to upper bound the watermark distortion:
>
> $$
> x_{\text{wm}} = x + \alpha_1 \times \tanh (\delta_{\theta}(x,m))
> $$
>
> We train until achieving good detection and decoding accuracy (95% mIoU and 100% bit accuracy). This serves as an initialization for the second phase. In the second phase, training is similar (using the same augmentations/masks for robustness/detection), but now the distortion is modulated by the JND:
>
> $$
> x_{\text{wm}} = x + \alpha_{\text{JND}} \cdot \mathrm{JND}(x) \odot \tanh(\delta_{\theta}(x,m))
> $$
>
> Without perceptual constraints, the PSNR would be lower, and the watermark is expected to be more robust. The balance is maintained by controlling the scalar value $\\alpha_{\text{JND}}\$, which can be adjusted during the second phase of training. Even if trained for a specific value (2 in our case), a different value can be used at inference to enhance robustness (albeit with increased visibility).
>
> - We illustrate this by showing the bit-accuracy of a 32-bit message for different values of $\\alpha_{\text{JND}}\$ (1, 1.5, 2.0, 2.5, and 3.0), against a combination of JPEG-80, center 50% crop + resize, and brightness 1.5
>
> | Method   | WAM_1.0 | WAM_1.5 | WAM_2.0 | WAM_2.5 | WAM_3.0 |
> |----------|---------|---------|---------|---------|---------|
> | PSNR     | 44.1    | 40.7    | 38.3    | 36.4    | 34.8    |
> | Bit Acc. | 71.0    | 84.3    | 88.8    | 90.8    | 91.9    |
>
> (Same set-up as table 2 on COCO). Thus, in practice, the WAM’s robustness/imperceptibility tradeoff can be controlled at inference time.

---

> > ### Author Response · Authors · 2024-11-19
> >
> > > 2) Could the authors discuss the trade-off between the bit capacity, robustness, imperceptibility? It is well-known that there is an inherent trade-off among these three dimensions in watermarking. In this paper, bit capacity was sacrificed, yet imperceptibility was not outstanding than the baseline, and robustness comparisons were limited to a specific scenario: localized watermarked regions.
> >
> > We clarified our operating point in the general comment. Additionally, we emphasize the following:
> >
> > - As shown in the None/Geometric/Valuemetric columns of Table 2, we demonstrate comparable or improved robustness not only for localized watermarked regions, as suggested by the reviewer, but also when the entire image is watermarked.
> >
> > - We offer three additional functionalities: localization, detection separate from decoding, and the ability to hide multiple 32-bit messages. The separation of detection from decoding allows the payload to be used solely for message hiding, which is a significant difference compared to other methods. The ability to hide multiple messages indicates a much higher true capacity, which implicitly enhances robustness.
> >
> > - The trade-off also arises from our focus on localization and robustness to small crops. WAM is robust in scenarios where only 10% of the image is cropped and resized, or when only 10% of the image is watermarked. For these cases—detailed in Table 7 of the appendix—WAM is significantly more robust than other methods.
> >
> > This is the operating point we have chosen intentionally, but a different one with less robustness (especially to geometric transformations) could have been selected.
> >
> >
> > > 3) While WAM used a JND map to enhance imperceptibility, its imperceptibility was comparable but not outstanding relative to baseline methods. Is there potential for further improvement in imperceptibility without compromising robustness?
> >
> > - WAM demonstrates comparable PSNR to other methods, but the JND's emphasis on the blue channel makes the watermark less visible to the human eye.
> > - As discussed in the future work section (Section 6), further improvements in imperceptibility might be achieved by employing more sophisticated Human Visual System (HVS) models or by regularizing the watermark to eliminate repetitive patterns during training. This approach could allow us to reduce PSNR, enhance robustness, and maintain similar imperceptibility.
> >
> > > 4) Given that the capacity of WAM was limited, its ability to support multiple distinct watermarks is constrained. As the amount of AIGC grows rapidly, do the authors have suggestions for extending WAM to support larger payload?
> >
> > **32 Bits is Sufficient:**
> > See limitations in Section 6. Unlike other methods that rely on decoding to verify the  presence of the watermark using some of the capacity, WAM separates detection from decoding, making 32 bits sufficient e.g.  for labeling each GenAI provider differently, and even if the watermarked zone is only 10% of the image.
> >
> > **How to Increase Capacity:**
> > Our focus is intentionally on 32 bits. However, for larger payloads:
> >
> > - Extending training (e.g., to 800 epochs) could increase capacity. We are testing this and will include findings in the camera-ready version.
> > - Increasing the localization constraint from 10% to 20% of 256x256, and/or diminishing the other robustness contrainsts
> > - Training on 512x512 resolution while maintaining a 10% target area could also increase capacity. Although more computationally intensive, this approach should handle larger payloads more effectively, as hiding n bits in 10% of a 512x512 image is easier than in 10% of a 256x256 image.

---

> > ### Comment · Reviewer_mbVM · 2024-11-21
> > **I would like the keep the rating as it is.**
> >
> > Thanks for the authors' feedback.
> >
> > The feedback partially addressed some of my previous comments. However, the key issue with the capacity and the complexity of the proposed method are not directly addressed with detailed and solid explanations and experiments. The authors' response may make some intuitive sense, but it is not solidly convincing.
> >
> > As a result, I will keep my rating as it is. And I intend to vote for rejection if there is no champion for strong acceptance.

---

> ### Author Response · Authors · 2024-11-22
>
> We respectfully disagree with the reviewer's statement regarding the capacity and complexity of our proposed method. We do not not only provide intuitive explanations but also detailed demonstrations throughout the paper and the rebuttal.
>
> **Capacity Concerns**:
>
> - We would like to reiterate that SSL does not hide 2048 bits, as previously mentioned by the reviewer, but rather 30 bits in the original paper, which is less than WAM. We also kindly invite the reviewer to refer to our response to reviewer n1mi, where we compare WAM to CIN, another watermarking technique that hides 30 bits, and WAM demonstrates greater robustness. The trade-off between capacity and robustness is a well-known challenge, and our method priorities robustness. This is our operating point: the importance is to be very robust, in which case we believe that 32 bits is enough
>
> 1) What application does the reviewer have in mind where 32 bits are insufficient? In the case of WAM, it is decoupled from detection, allowing the messages to be fully utilized to hide, for example, each ~user's ID or~ AI tool ID, for which 32 bits are enough ~(approximately 4 billion different IDs!)~ if robust*.
> 2) Could the reviewer point to any watermarking method with a higher capacity that is more robust than WAM, particularly in evaluations similar to Table 2, where we highlight WAM's robustness against geometric transformations?
>
> **Complexity Concerns**
>
> - We have addressed the complexity of our method both in the paper (limitations paragraph of section 6) and in our rebuttal by adding experiments comparing the time required for embedding and extracting watermarks with other methods. Could the reviewer specify what additional experiments they were waiting for?
>
> We appreciate the reviewer's feedback and are open to further discussion to clarify any remaining concerns.
>
> *We crossed "user's ID" and "approx 4B different IDs" present in the original version of this answer as the first part was a typo and the second did not convey the correct message. As discussed in our next comment,  our paper's motivation is about the "application scenario of AIGC", which is not about tracing users, but about detecting AI-generated images, such as attributing a message per AI generator, not per user.

---

> ### Comment · Reviewer_mbVM · 2024-11-22
> **What is the maximum number of unique messages that is used in the experiments of the paper?**
>
> The capacity of an image watermarking system is often misunderstood as the number of bits of a message. However, the true meaning of the capacity of a watermarking system is the total number of unique messages that can be supported by a watermarking system without causing significant confusion between different messages when detecting watermarks.
>
> This means, even if you use 32 bits for a message, the number of non-confusing messages, which is the true capacity of the watermarking system, is often much smaller than 2^32, and it will be even smaller if the images are under attacks.
>
> The reviewer's question is related to the true capacity of the proposed watermarking system. Not the theoretical upper-bound given by 2^32.
>
> More often than not, the true capacity is difficult to derive theoretically. One way to demonstrate a high true capacity is to sample a large number of unique messages and evaluate false positive rates on a large enough dataset (typically starting from the scale of 1e6).
>
> To conclude, the final question boils down to how many unique messages are used in the experiments? and how many unique images are used?
>
> If any of the numbers are small, then the experimental results are not statistically significant, thus not convincing.

---

> > ### Author Response · Authors · 2024-11-23
> >
> > >If any of the numbers are small, then the experimental results are not statistically significant, thus not convincing.
> >
> > Interesting thoughts. Please, would you mind citing one paper where you have been convinced about the experimental results because *these numbers* were large enough?
> >
> > We are very eager to learn how high are your standards in watermarking.

---

> > > ### Comment · Reviewer_mbVM · 2024-11-23
> > > **The authors' feedback did not directly answer the reviewer's previous questions.**
> > >
> > > The reviewer's question is about the statistical significance of the experiments in demonstrating the capacity of the proposed image watermarking system.
> > >
> > > A large capacity for a watermarking system is an important practical requirement, because it determines the maximum number of unique users who can be served by the watermarking system simultaneously, as each user is associated with a unique message. It is possible that a watermarking system performs well when there are only 100 unique users, but the performance may drop significantly when the number of users is large. An image watermarking system with small capacity is not quite useful in practice, especially in the application scenario of AIGC, where hundreds of thousands of users generate huge amount of images.
> > >
> > > An image watermarking system may work quite well when there are 100 users, but its performance may drop significantly when the number of unique users increases at scale. Please justify the statistical significance of the experimental results in demonstrating that the proposed watermarking system has a high capacity.
> > >
> > > Since the authors' previous response did not address the reviewer's question on the capacity of the image watermarking system, the reviewer decides to temporarily lower the rating until this question is addressed.

---

> ### Author Response · Authors · 2024-11-23
>
> >An image watermarking system with small capacity is not quite useful in practice, especially in the application scenario of AIGC, where hundreds of thousands of users generate huge amount of images.
>
> Our paper's motivation is indeed about the "*application scenario of AIGC*", which is not about tracing users, but about **detecting AI-generated images**. We kindly invite the reviewer to have a look at the regulations we cited in the first paragraph. None of them requires to trace users. On the contrary, this would be an infringement of the GDPR Privacy Act in Europe.
>
> The application we are targeting is more about attributing a message per AI generator, so that even when splicing images generated by several generators, we can still localize and attribute each part of the image to a generator.
>
> As for the capacity, we can of course rely to Shannon theory applied to Binary Symmetric Channel. The capacity is $1-H(p)$ where $H$ is the binary entropy and $p$ the Bit Error Rate. This means that we can transmit $n(1-H(p))$ bits addressing $2^{n(1-H(p))}$ generators.
>
> But all of this is misleading in practice. Asking for the *capacity of the image watermarking scheme* does not make sense in practice because the Bit Error Rate $p$ goes to $1/2$ as one distorts more and more the images. In real life, there is no such thing as a maximum distortion giving birth to a maximum $p$ and thus, a would-be guaranteed capacity. This is the humble point of view of a practitioner.
>
> > Since the authors' previous response did not address the reviewer's question on the capacity of the image watermarking system, the reviewer decides to temporarily lower the rating until this question is addressed
>
> The question that the reviewer is referring to is "What is the maximum number of unique messages that is used in the experiments of the paper?". For table 2, as detailed in line 425, we evaluate the detection rates and bit accuracies on 10000 images of the COCO validation set. It is done with a different randomly generated 32-bit message for each image.
>
> Is it what the reviewer is asking for? Could the reviewer please also provide one or a list of "*convincing*" papers according to the reviewer's standards, with large numbers of unique messages and unique images?

---

> ### Author Response · Authors · 2024-11-24
>
> Another way to approach the problem is to consider it as a statistical detection test.
> We consider the null hypothesis $H_0$ that each bit of the output binary message $\hat{m}$ is independent and distributed as a Bernoulli variable with probability of success $0.5$, and the alternative hypothesis $H_1$ which is that $\hat{m} = m$.
> Given an observed bit accuracy $bit.acc.(m, \hat{m})$,
> the $p$-value is the probability of observing a bit accuracy at least as extreme as the one obtained under the null hypothesis.
> It is given by the cumulative distribution function of the binomial distribution:
> $$
> pvalue(m, \hat{m}) = \sum_{k \geq p }^{n} \binom{n}{k} \frac{1}{2^{n}} = I_{1/2} \left( n \cdot p, n \cdot (1 - p) + 1 \right)
> $$
>
> where
> - $n$ **is the number of bits**,
> - $ p = bit.acc.(m, \hat{m}) $,
> - the c.d.f. of the binomial is expressed by $I_x(a, b)$, the regularized incomplete Beta function.
>
>
>
> To further validate this, we additionally compute the bit accuracy and the $-log_{10}$ p-value on 100 images of COCO, and report the average in the following table.
> What we can observe is that, indeed, MBRS and TrustMark have higher capacity, which results in better $-log_{10}$ p-value for many augmentations (identity, contrast, jpeg). However, there are less robust: for instance for a combined augmentation of JPEG60 Crop 50% (in area) and brightness 1.5 no method achieve a $-log_{10}$ p-value bigger than $1$, while ours is bigger than $5$ (note that is is without using the detection map given by WAM, only the extracted bits).
>
> | Model       | $n$ | bit acc Identity | - log pvalue Identity | bit acc Contrast 1.5 | - log pvalue Contrast 1.5 | bit acc JPEG 50 | - log pvalue JPEG 50 | bit acc Rotate 5 | - log pvalue Rotate 5 | bit acc HorizontalFlip | - log pvalue HorizontalFlip | bit acc Crop 0.71 | - log pvalue Crop 0.71 | bit acc Perspective 0.1 | - log pvalue Perspective 0.1 | bit acc (JPEG Crop Brightness) (60 0.71 0.5) | - log pvalue (JPEG Crop Brightness) (60 0.71 0.5) |
> |-------------|-----|------------------|---------------------|----------------------|-------------------------|-----------------|--------------------|------------------|---------------------|------------------------|---------------------------|-------------------|----------------------|-------------------------|----------------------------|----------------------------------------------|-------------------------------------------------|
> | cin         |  30 | **1.00**         | 9.03                | **1.00**             | 9.03                    | **1.00**        | 9.03               | 0.48             | 0.31                | 0.51                   | 0.42                      | 0.52              | 0.42                 | 0.48                    | 0.41                       | 0.51                                      | 0.39                                            |
> | mbrs        | 256 | 0.97             | **62.30**           | 0.88                 | **39.52**                   | 0.96            | **61.77**              | 0.50             | 0.37                | 0.50                   | 0.48                      | 0.50              | 0.44                 | 0.50                    | 0.39                       | 0.50                                         | 0.41                                            |
> | trustmark   | 100 | **1.00**         | 29.87               | 0.96                 | 24.60                   | **1.00**        | 29.91              | 0.60             | 2.70                | **1.00**               | **29.90**                     | 0.56              | 1.21                 | 0.92                    | **20.45**                  | 0.53                                         | 0.91                                            |
> | wam         |  32 | **1.00**         | 9.63                | **1.00**             | 9.63                | **1.00**        | 9.63           | **0.97**         | **8.49**            | **1.00**               | 9.63                  | **0.98**          | **9.01**             | **0.99**                | 9.28                   | **0.84**                                         | **5.60**                                        |
>
>
> One could also extend the previous approach in the case of multiple messages.
> We compare the message $\hat m$ from the watermark extractor to $ m_{1}, . . . , m_{N'}$ (where $N'$ would be the number of messages, representing for instance $N'$ generative models). If the $N'$ hypotheses are rejected, we can conclude that the image was not generated by any of the models. With regards to the detection task, false positives ( i.e., obtaining a collision with one of the message by chance) are more likely since there are N' tests.
> The new p-value would be $pvalue_{N'} (m , \hat m) = 1 - (1 - pvalue(m, \hat m))^{N'} \approx N' pvalue(m, \hat m)$.
> We can report these results if needed.

---

> > ### Comment · Reviewer_mbVM · 2024-11-27
> > **Thanks for the detailed answer of the authors.**
> >
> > I really appreciate the authors detailed answers to my previous questions. However, I am not convinced on the following points:
> >
> > 1. Even for detection purpose, the capacity of a watermarking system is also very important. As far as I know, many industry products of watermark detection service value the capacity, because it determines how many clients their product could serve. For detection purpose, it is often a business run in B2B mode, where the clients are companies instead of individual persons. However, the capacity still matters a lot.
> >
> > 2. Thanks for clarifying that the experiments uses 10,000 images, and each with a different randomly generated message. However, the number of images used is not matching the SOTA experimental setting in this field. SSL uses 1e6 images to check for false positives. Please note that false positive rate is closely related to the capacity of a watermarking system. If a system uses more messages at the same time to watermark different images, then the chance to have more false positive cases is often larger. This paper does not systematically study this important perspective of performance.
> >
> > In summary, I would still like to keep my initial rating as marginally below acceptance. And I am leaning towards rejection unless there is a champion for acceptance.

---

> > > ### Author Response · Authors · 2024-11-28
> > >
> > > We thank the reviewer for their feedback.
> > >
> > > > 1 [...] However, the capacity still matters a lot.
> > >
> > > We fully agree that the ability to hide more bits would be great, and we have acknowledged this as a limitation in our paper. However, we believe that our work already provides significant contributions, such as localization, separate detection from decoding, and the capability to hide multiple messages. Additionally, 32 bits is not super low and is sufficient for many applications.
> > >
> > > In response to the reviewer's suggestion, we evaluated WAM on the 1.2+ million ImageNet (non-watermarked) training set.
> > >
> > > > 2 [...] If a system uses more messages at the same time to watermark different images, then the chance to have more false positive cases is often larger. This paper does not systematically study this important perspective of performance.
> > >
> > > We would like to remind the reviewer that WAM works differently than other methods as it provides a detection separate from decoding. The extractor provides a detection bit for each pixel, and an image is flagged as watermarked if at least a certain proportion of its pixels are detected as watermarked (see Line 162). For ImageNet, the different false positive rates (FPRs) are as follows:
> > >
> > > | Threshold  | > 0.05  | > 0.07  | > 0.1   | > 0.2   | > 0.3   |
> > > |------------|---------|---------|---------|---------|---------|
> > > | FPR        | 2.01e-3 | 1.52e-3 | 6.89e-4 | 1.24e-4 | 3.67e-5 |
> > >
> > > Subsequently, a message is extracted from the detected pixels, followed by a majority vote (Line 172). We observe empirically that the average output for each of the 32 bits is random (on non watermarked images that have at least one pixel falsely detected as watermarked). The output message of WAM on a non-watermarked image falsely flagged as watermarked will therefore be random. So in scenarios where WAM watermarks content for different companies, the probability to match a specific company's message on a non watermarked image flagged as watermarked can be easily computed if a threshold is given on the bit error rate. **Overall, in the setting suggested by the reviewer, the first detection phase serves as an effective filter to limit the FPR, even before the second phase, which is common to other methods (but for which indeed more bits would be better).**
> > >
> > > The other false positive case is if an image wm with message m is flagged as being wm with message m’. This case is strictly equivalent to SSL (which hides 30 bits in the paper). We agree that we have only proved superior robustness in this setting on 10K validation images from COCO, we can include these comparisons on 1M+ images in the manuscript, as suggested by the reviewer.
> > >
> > > We regret that our choice of operating point, or evaluating on 10k instead of 1M, could be a reason for rejection. We hope to have addressed the reviewer's concerns and demonstrated the value of our contributions.

---

### Official Review · Reviewer_p14G · 2024-11-05

**Soundness:** 3
**Presentation:** 3
**Contribution:** 3
**Rating:** 8
**Confidence:** 3

**Summary:**

This paper proposes a new method, Watermark Anything Model (WAM), for adding watermarks to images. The method demonstrates state-of-the-art performance even under various manipulations, such as splicing, geometric transformations, color changes, and inpainting. It consists of an embedder and an extractor capable of localizing the watermark region and decoding distinct messages in different areas of the image. The paper is well-written and easy to understand. Overall, the proposed approach offers a robust solution for image watermarking, with potential applications across diverse digital media fields.

**Strengths:**

The method has been tested against various manipulations, showcasing its robustness. It reliably detects watermarks even in image splicing and inpainting, including small or detailed areas. The paper is well-structured, making it clear, easy to read, and accessible to the reader.

**Weaknesses:**

The watermark model has not yet been tested against attacks from generative AI, which is widely used today. The paper would be strengthened if the watermark remains detectable after processing with super-resolution models (whether CNN-based or Stable Diffusion-based). Otherwise, people could easily remove the watermark without much effort. The model demonstrates a 100% true positive rate in many cases; however, most of these manipulations are included in the training augmentation, which may indicate overfitting rather than true robustness. If a manipulation is not included in the training data, can the watermark still be detected? Please also elaborate on the model performance against VAE attack and diffusion attack.

**Questions:**

In the paper, the authors talk about generative AI models, but they do not show how strong the model is against these types of AI, like Stable Diffusion, which is popular in the AI art community. With Stable Diffusion, it’s possible that all the hidden messages could disappear when the image is changed. It’s important to show that the model can handle generative AI modifications, like Stable Diffusion. For example, if I use Stable Diffusion-based super-resolution (a common tool for improving image quality) on the image, can the model still detect the watermark? If not, the watermark becomes easy to remove.

Another useful test would be to see if adding a second watermark to an image removes or covers the first one. For instance, after editing an image, if I use WAM again to add my own watermark, can the original watermark message still be found?

Please do these tests to show how strong the model is, or suggest ways to prevent these possible problems.

---

> ### Author Response · Authors · 2024-11-19
>
> We thank the reviewer for their valuable feedback and time spent on our paper.
>
> > The model demonstrates a 100% true positive rate in many cases; however, most of these manipulations are included in the training augmentation, which may indicate overfitting rather than true robustness. If a manipulation is not included in the training data, can the watermark still be detected? Please also elaborate on the model performance against VAE attack and diffusion attack.
>
> - We would like to emphasize that the primary focus of our paper is not solely on robustness. Our main contribution lies in introducing three novel capabilities: localization, multiple watermarks, and detection separate from decoding, while maintaining comparable robustness.
>
> - For robustness, our goal is to achieve extreme resilience against typical internet/social network image sharing, which is why we incorporate extensive augmentation during training. It is not trivial overfitting: it is challenging to maintain robustness across all augmentations.  Target ones can be integrated into the training pipeline to for specific robustness.
>
> - Regarding the VAE attack, we conducted new experiments to assess WAM's performance. We measured bit accuracy for a 32-bit message (consistent with Table 2 and as detailed in our response to Reviewer oRgo for MBRS and CIN newly added). WAM remains robust until the VAE significantly compresses the image (PSNR 25). In this scenario, only MBRS and Trustmark maintain robustness, but they lack resilience against geometric transformations, which represents a different trade-off choice. We will include these findings in the paper.
>
>
> | Model                               | HiDDeN | WAM  | DWTDCT | SSL  | FNNS | TrustMark | MBRS |CIN |
> |-------------------------------------|--------|------|--------|------|------|-----------|------|------|
> | bmshj2018_hyperprior (PSNR 32.9)      | 82.8   | 100.0| 72.4   | 98.4 | 97.9 | 100.0     |100.0 | 98.1      |
> | bmshj2018_fact_8 (PSNR 32.4)          | 80.6   | 99.8 | 63.1   | 97.6 | 95.1 | 100.0     |100.0 | 100.0      |
> | bmshj2018_fact_4 (PSNR 28.6)          | 56.9   | 98.6 | 49.7   | 66.8 | 55.9 | 99.9      |100.0 | 99.7      |
> | bmshj2018_fact_1 (PSNR 25.5)          | 50.7   | 53.7 | 49.5   | 51.6 | 50.2 | 98.1      |97.7 | 67.6      |
>
>
> > The watermark model has not yet been tested against attacks from generative AI, which is widely used today. [...] . It’s important to show that the model can handle generative AI modifications, like Stable Diffusion..
>
> - As shown in Table 2 and Figure 10 in the appendix, we demonstrate WAM's robustness against heavy inpainting. WAM is not only robust to such modifications but can also precisely localize which subparts have been edited
> - Regarding full-image purification via Stable Diffusion, WAM shares the same limitations as other methods. We acknowledge that this is an important research area, but it is beyond the current scope of our work.
> - We provide a new comparison focusing on decoding 32 bits using the DiffPure method and implementation from [1]. We observe that WAM shows comparable robustness and breaks when the purification is heavy. We will include these findings in the paper.
>
>
>
> | Metric                | HiDDeN | WAM  | DWTDCT | SSL  | FNNS | TrustMark | MBRS | CIN  | WAM  |
> |-----------------------|--------|------|--------|------|------|-----------|------|------|------|
> | DiffPure_0.001 (PSNR 30.1) | 88.6   | 100  | 80.4   | 99.2 | 97.9 | 100.0     | 100.0| 100.0| 100.0|
> | DiffPure_005 (PSNR 28.4)   | 65.2   | 99.4 | 48.6   | 63.0 | 64.1 | 98.3      | 99.5 | 100.0| 99.4 |
> | DiffPure_01 (PSNR 27.0)    | 56.2   | 71.3 | 49.0   | 54.2 | 53.0 | 74.9      | 85.1 | 82.3 | 71.3 |
> | DiffPure_02 (PSNR 24.9)    | 51.6   | 49.1 | 49.5   | 52.0 | 50.1 | 54.3      | 64.8 | 61.9 | 49.1 |
>
>
> > Another useful test would be to see if adding a second watermark to an image removes or covers the first one. For instance, after editing an image, if I use WAM again to add my own watermark, can the original watermark message still be found?
>
> We demonstrate that, on average across 100 images, WAM can still detect the presence of a watermark even if two watermarks overlap entirely on the whole image with different messages. We appreciate the reviewer for raising this point, as it highlights the advantage of having decoding separated from detection.
>
> |                        | One Watermark | Two Overlapping Watermarks |
> |--------------------|---------------|----------------------------|
> | Detection Accuracy | 100%          | 100%                        |
>
> - However, WAM will not be able to decode both messages if one watermark is directly on top of the other. In practice, if the detector and encoder are managed by the same person, they can verify that the image is not already watermarked by WAM before adding a second watermark.
>
>
> [1] Saberi et al., 2024, Robustness of AI-Image Detectors: Fundamental Limits and Practical Attacks, ICLR 2024

---

> > ### Author Response · Authors · 2024-11-27
> >
> > Dear Reviewer,
> >
> > We wanted to follow up to ensure that our responses to your comments were satisfactory. Please feel free to reach out if you require any further clarifications. And thank you again for your time!

---

> > > ### Author Response · Authors · 2024-12-02
> > >
> > > Dear Reviewer,
> > >
> > > Today is the final day for reviewers' comment posting. We would appreciate updates you can provide.
> > >
> > > Thank you!
> > >
> > > The Authors

---

### Author Response · Authors · 2024-11-19
**General Comment to AC and reviewers**

We appreciate the feedback from all six reviewers on our paper.

**Operating Point: Intentional 32-bit Capacity of WAM**

This is discussed in the limitations section. More specifically:

- Our goal is to achieve extreme robustness against typical internet/social network image sharing, which necessitates a trade-off with capacity. WAM is particularly robust against "Geometric augmentations" compared to other methods (see Table 2). It can localize watermarks and extract a 32-bit message from just 10% of a 256x256 image, which is the operating point we have chosen; another one could have been chosen.
- WAM offers three additional functionalities: localization, the ability to hide multiple 32-bit messages, and detection separate from decoding. The second functionality indicates a higher true capacity, and the third ensures that 32 bits are sufficient for many applications.

**Additional Experiments**
- We have added comparisons* with CIN [2] and MBRS [1], which do not provide localization. While these methods are robust against value metric transformations, they lack robustness against geometric augmentations, representing a different operating point (as requested by reviewer oRgo, see official response for details).

| Method | None          | Geometric     | ValueMetric   | Splicing      | PSNR/SSIM/LPIPS |
|--------|---------------|---------------|---------------|---------------|-----------------|
| MBRS   | 100.0 | 50.4  | 99.8 | 65.31  | 38.8/0.99/0.07 |
| CIN    | 100.0 | 50.3  | 100.0 | 96.47 | 38.3/0.99/0.08 |
| WAM    | 100.0 | 91.8  | 100.0 | 95.3 | 38.3/0.99/0.04 |

- Speed tests demonstrate WAM's competitive performance (requested by reviewers oRgo and n1mi, see responses for details).

| Method         | HiDDeN | WAM  | DWTDTC | SSL   | FNNS  | TrustMark | MBRS  | CIN   |
|----------------|--------|------|--------|-------|-------|-----------|-------|-------|
| Encoding Time  | 0.070  | 0.040| 0.041  | 0.228 | 0.508 | 0.026     | 0.052 | 0.124 |
| Decoding Time  | 0.026  | 0.032| 0.033  | 0.017 | 0.024 | 0.022     | 0.028 | 0.074 |

- We evaluated robustness against diffusion-based purification, acknowledging its challenge for all watermarks (requested by reviewers p14G and ITj6, see responses for details). We agree that AI-based purification is a significant challenge for image watermarks, but it is not addressed in our work, and we share the same limitations as others. However, Table 2 shows WAM's robustness against strong inpainting.

| Metric                | HiDDeN | WAM  | DWTDCT | SSL  | FNNS | TrustMark | MBRS | CIN  | WAM  |
|-----------------------|--------|------|--------|------|------|-----------|------|------|------|
| DiffPure_0.001 (PSNR 30.1) | 88.6   | 100  | 80.4   | 99.2 | 97.9 | 100.0     | 100.0| 100.0| 100.0|
| DiffPure_005 (PSNR 28.4)   | 65.2   | 99.4 | 48.6   | 63.0 | 64.1 | 98.3      | 99.5 | 100.0| 99.4 |
| DiffPure_01 (PSNR 27.0)    | 56.2   | 71.3 | 49.0   | 54.2 | 53.0 | 74.9      | 85.1 | 82.3 | 71.3 |
| DiffPure_02 (PSNR 24.9)    | 51.6   | 49.1 | 49.5   | 52.0 | 50.1 | 54.3      | 64.8 | 61.9 | 49.1 |

- WAM exhibits good robustness against VAE attacks, comparable to other methods (see response to reviewer p14G).
- We can control the robustness/imperceptibility trade-off at encoding time by varying the distortion factor (see experiments in response to reviewer mbVM).
- We can still detect if one watermark is added on top of another (see p14G and ITj6)

**Upcoming Experiments for Camera Ready**

These experiments require full retraining:

- We will compare WAM's performance with different extractor sizes, as the reviewers have asked for more details and discussion about that. We agree that it would be nice to have smaller extractors especially, as stated in the limitation paragraph of section 6.
- We will explore WAM with higher capacity by reducing robustness requirements and/or longer training.

Finally, two reviewers mentioned the paper MuST [3]. We agree that it is relevant, but the approach is very different: it decodes different parts of the image independently while WAM uses a single extractor that automatically performs segmentation (see response to pZCp). We will cite it, but note that no model or training script is available for reproducibility. We also remind the AC and reviewers that we commit to open-sourcing models and code upon publication.

*we revised the table because of a typo

[1] Fang H, Qiu Y, Chen K, et al. Flow-based robust watermarking with invertible noise layer for black-box distortions[C]//Proceedings of the AAAI conference on artificial intelligence. 2023

[2] Ma R, Guo M, Hou Y, et al. Towards blind watermarking: Combining invertible and non-invertible mechanisms[C]//Proceedings of the 30th ACM International Conference on Multimedia. 2022: 1532-1542.

[3] Wang G, Ma Z, Liu C, et al. MuST: Robust Image Watermarking for Multi-Source Tracing[C]//Proceedings of the AAAI Conference on Artificial Intelligence. 2024, 38(6): 5364-5371.

---

### Comment · Area_Chair_4Qv5 · 2024-11-25

Hi Reviewers,

We are approaching the deadline for author-reviewer discussion phase. Authors has already provided their rebuttal. In case you haven't checked them, please look at them ASAP. Thanks a million for your help!

---

### Meta-Review · Area_Chair_4Qv5 · 2024-12-20

**Metareview:**

This paper works on watermarking. Authors proposed a deep-learning model for localized image watermarking, WAM, which could extract different watermarks from different regions.

This paper was reviewed by 6 reviewers and got mixed scores as three 8, two 5, one 3.

Strength and weaknesses given by reviewers before rebuttals are as follows (notes that different reviewers has different perspectives of the paper, so conflicts in strength and weaknesses might happen):

The strength of the papers are: 1) paper is well written; 2) proposed method is novel; 3) results are comprehensive.

Weaknesses are: 1) model is not tested against attack from generative AI and generalization capability is not tested; 2) capacity is limited to 32 bits. some comparison is not fair; 3) lacks discussion of computational complexity; 4) embedding and extracting multiple watermarks in different regions was explored before; 5) need more experiments; 6) novelty is limited.

During rebuttal, Reviewer p14G (rating 8) suggested that their concerns are well addressed by authors.

Reviewer mbVM (rating 5) suggested that their concerns on capacity are not addressed well. but also mentioned that "And I am leaning towards rejection unless there is a champion for acceptance.".

Reviewer n1mi (rating 5) didn't give comments in the rebuttal discussion.

Reviewer oRgo (rating 8) said they are positive about this paper and maintained the score. But suggested the experimental setup is incomplete and some of the conclusions are not solid.

Reviewer pZCp (rating 8) gave positive views about this paper. but also suggested several ways to improve the paper.

Reviewer UTj6 (rating 3) thought authors addressed their concerns and would like to raise the score. But at the end mentioned "The rebuttal period has been extended, but the authors have not updated the PDF as they promised." AC believed there was some miscommunication here.

Given all these discussions, AC found in general reviewers were positive about this paper and didn't have strong opinions to reject this paper. So AC decided to accept this paper.

**Additional Comments On Reviewer Discussion:**

During rebuttal, Reviewer p14G (rating 8) suggested that their concerns are well addressed by authors.

Reviewer mbVM (rating 5) suggested that their concerns on capacity are not addressed well. but also mentioned that "And I am leaning towards rejection unless there is a champion for acceptance.".

Reviewer n1mi (rating 5) didn't give comments in the rebuttal discussion.

Reviewer oRgo (rating 8) said they are positive about this paper and maintained the score. But suggested the experimental setup is incomplete and some of the conclusions are not solid.

Reviewer pZCp (rating 8) gave positive views about this paper. but also suggested several ways to improve the paper.

Reviewer UTj6 (rating 3) thought authors addressed their concerns and would like to raise the score. But at the end mentioned "The rebuttal period has been extended, but the authors have not updated the PDF as they promised." AC believed there was some miscommunication here.

Given all these discussions, AC found in general reviewers were positive about this paper and didn't have strong opinions to reject this paper. So AC decided to accept this paper.

---

### Decision · Program_Chairs · 2025-01-22

Accept (Poster)

---

> ### Public Comment · ~Tom_Sander1 · 2025-02-28
> **PDF updated**
>
> Thank you. We updated the camera ready submission, incorporating the rebuttals discussion.